# Identification of PARP-7 substrates reveals a role for MARylation in microtubule control in ovarian cancer cells

Lavanya H Palavalli Parsons[1,2,3†‡], Sridevi Challa[1,2†], Bryan A Gibson[1,2†§], Tulip Nandu[1,2], MiKayla S Stokes[1,2], Dan Huang[1,2,4], Jayanthi S Lea[3], W Lee Kraus[1,2*]

[1]Laboratory of Signaling and Gene Regulation, Cecil H. and Ida Green Center for Reproductive Biology Sciences, University of Texas Southwestern Medical Center, Dallas, United States; [2]Division of Basic Research, Department of Obstetrics and Gynecology, University of Texas Southwestern Medical Center, Dallas, United States; [3]Division of Gynecologic Oncology, Department of Obstetrics and Gynecology, University of Texas Southwestern Medical Center, Dallas, United States; [4]Department of Cardiology, Clinical Center for Human Gene Research, Union Hospital, Tongji Medical College, Huazhong University of Science and Technology, Wuhan, China

*For correspondence:
LEE.KRAUS@utsouthwestern.edu

†These authors contributed equally to this work

Present address: ‡Department of Obstetrics, Gynecology, and Reproductive Sciences, University of Texas Health Science Center at Houston, McGovern Medical School, Houston, United States; §Department of Biophysics, University of Texas Southwestern Medical Center, Dallas, United States

**Abstract** PARP-7 (TiPARP) is a mono(ADP-ribosyl) transferase whose protein substrates and biological activities are poorly understood. We observed that *PARP7* mRNA levels are lower in ovarian cancer patient samples compared to non-cancerous tissue, but PARP-7 protein nonetheless contributes to several cancer-related biological endpoints in ovarian cancer cells (e.g. growth, migration). Global gene expression analyses in ovarian cancer cells subjected to PARP-7 depletion indicate biological roles for PARP-7 in cell-cell adhesion and gene regulation. To identify the MARylated substrates of PARP-7 in ovarian cancer cells, we developed an NAD$^+$ analog-sensitive approach, which we coupled with mass spectrometry to identify the PARP-7 ADP-ribosylated proteome in ovarian cancer cells, including cell-cell adhesion and cytoskeletal proteins. Specifically, we found that PARP-7 MARylates α-tubulin to promote microtubule instability, which may regulate ovarian cancer cell growth and motility. In sum, we identified an extensive PARP-7 ADP-ribosylated proteome with important roles in cancer-related cellular phenotypes.

## Introduction

Members of the poly(ADP-ribose) polymerase (PARP) family of enzymes catalyze ADP-ribosylation (ADPRylation), a posttranslational modification of proteins, through covalent transfer of ADP-ribose (ADPR) from β-nicotinamide adenine dinucleotide (NAD$^+$) onto a variety of amino acid residues, including glutamate, aspartate, and serine (*Gibson and Kraus, 2012*; *Gupte et al., 2017*). ADPRylation can occur as a single ADP-ribose (i.e. mono(ADP-ribose), [MAR]) or multiple ADP-ribose moieties (i.e. poly(ADP-ribose), [PAR]) (*Gibson and Kraus, 2012*; *O'Sullivan et al., 2019*). Both MAR and PAR modifications are reversible and can be removed by a variety of ADPR hydrolases (*O'Sullivan et al., 2019*; *Rack et al., 2020*). Free and protein-linked ADP-ribose moieties can be bound by proteins ('readers') containing ADPR-binding domains (e.g. macrodomains, WWE domains), allowing MAR or PAR to be interpreted or 'read' (*Gibson and Kraus, 2012*; *Teloni and Altmeyer, 2016*). ADPRylation can modulate target protein functions, including enzymatic activity, interactions with binding partners, and protein stability through both direct effects resulting from

**eLife digest** Cancer is a complex illness where changes inside healthy cells causes them to grow and reproduce rapidly. Specialized proteins called enzymes – which regulate chemical reactions in the cell – often help cancer develop and spread through the body. One such enzyme called PARP-7 labels other proteins by attaching a chemical group which changes their behavior. However, it was unknown which proteins PARP-7 modifies and how this tag alters the actions of these proteins.

To investigate this, Parsons, Challa, Gibson et al. developed a method to find and identify the proteins labelled by PARP-7 in ovarian cancer cells taken from patients and cultured in the laboratory. This revealed that PARP-7 labels hundreds of different proteins, including adhesion proteins which affect the connections between cells and cytoskeletal proteins which regulate a cell's shape and how it moves.

One of the cytoskeletal proteins modified by PARP-7 is α-tubulin, which joins together with other tubulins to form long, tube-like structures known as microtubules. Parsons et al. found that when α-tubulin is labelled by PARP-7, it creates unstable microtubules that alter how the cancer cells grow and move. They discovered that depleting PARP-7 or mutating the sites where it modifies α-tubulin increased the stability of microtubules and slowed the growth of ovarian cancer cells.

Ovarian cancer is the fifth leading cause of cancer-related deaths among women in the United States. A new drug which suppresses the activity of PARP-7 has recently been developed, and this drug could potentially be used to treat ovarian cancer patients with high levels of PARP-7. Clinical trials are ongoing to see how this drug affects the behavior of cancer cells in patients.

chemical modification and indirect effects mediated by ADPR-binding 'reader' proteins (*Gupte et al., 2017*; *Kim et al., 2020*; *Ryu et al., 2015*).

In humans, the PARP family includes 17 members, each with unique structural domains, expression patterns, targets, enzymatic activity, localization, and functions (*Amé et al., 2004*; *Vyas et al., 2013*; *Vyas et al., 2014*). In spite of the moniker, PARP family members are mostly monoADPR transferases (MARTs). PARP family MARTs include PARPs 3, 4, 6–12, and 14–16 (*Vyas et al., 2014*). Much of the research on the PARP family has focused on its founding member, PARP-1, which plays roles in transcription and DNA damage repair (*Pascal, 2018*; *Ray Chaudhuri and Nussenzweig, 2017*), but the molecular and cellular functions of the PARP family are considerably more diverse (*Gibson and Kraus, 2012*; *Vyas et al., 2013*). Recent efforts to understand how ADP-ribosylation by other PARPs modulates cellular biology have identified roles in stress responses, cellular metabolism, and immune function (*Gupte et al., 2017*; *Kim et al., 2020*; *Luo and Kraus, 2011*; *Luo and Kraus, 2012*; *Ryu et al., 2015*). Further understanding of the diversity of functions of PARP family members will require more information about the catalytic activities of these enzymes, as well as identification of their substrate proteins. In this study, we addressed these questions for PARP-7 through an examination of its catalytic activity and identification of its substrates.

PARP-7 is a MART that localizes to both the cytoplasm and nucleus (*MacPherson et al., 2013*; *Vyas et al., 2014*). PARP-7 is also known as 2,3,7,8-tetrachlorodibenzo-p-dioxin (TCDD)-inducible PARP (TiPARP) because it is induced by TCDD as a target gene of the aryl hydrocarbon receptor (AHR) (*Ma et al., 2001*). PARP-7 contains a carboxyl-terminal catalytic domain, as well as central region containing a WWE (tryptophan-tryptophan-glutamate) motif thought to bind PAR and a single CCCH-type zinc-finger (*Aravind, 2001*; *Schreiber et al., 2006*; *Wang et al., 2012*). Unlike true PARPs, which contain a signature H-Y-E motif in their catalytic domains, PARP-7 contains H-Y-I (H532, Y564, and I631), which underlies its MART activity (*MacPherson et al., 2013*; *Vyas et al., 2014*). The histidine and tyrosine residues of the H-Y-E catalytic triad are required for the binding of NAD$^+$ in the ADP-ribosyl transferase catalytic site, whereas the glutamate residue is required for catalysis of PARylation activity (*Marsischky et al., 1995*; *Papini et al., 1989*). Little is known about the catalytic activity and substrates of PARP-7, although previous studies have identified a few targets, including PARP-7 itself through automodification, as well as histones, AHR, and liver X receptors (LXRs) (*Bindesbøll et al., 2016*; *Gomez et al., 2018*; *Ma et al., 2001*; *MacPherson et al., 2013*). *Gomez et al., 2018* showed that the sites of PARP-7 automodification are resistant to meta-iodobenzylguanidine (MIBG), an inhibitor of arginine-specific MARTs (*Loesberg et al., 1990*), but

are sensitive to iodoacetamide and hydroxylamine (*Zhang et al., 2013*), implicating cysteines and acidic side chain residues (e.g. glutamate and aspartate), respectively, as target residues for MARylation. More work is needed in this area, with higher throughput substrate identification.

Although some progress has been made, the biology of PARP-7 is poorly characterized and limited in scope. PARP-7 is involved in a negative feedback loop regulating AHR responses (*MacPherson et al., 2013*), but also plays a role in other cellular processes, including stem cell pluripotency, transcriptional regulation, mitotic spindle formation, viral replication, immune responses, and neuronal function (*Grimaldi et al., 2019*; *Kozaki et al., 2017*; *Roper et al., 2014*; *Vyas et al., 2013*; *Yamada et al., 2016*). PARP-7 has recently emerged as a potentially interesting target for cancer therapy, in part because breast cancers have altered expression of *PARP7* mRNA (*Cheng et al., 2019*). In addition to studies in breast cancer, the *PARP7* gene (located at 3q25) was identified in a susceptibility locus for ovarian cancer in a genome-wide association study (*Goode et al., 2010*). Determining its role in cellular signaling remains an important step in determining the utility of PARP-7 as a therapeutic target for breast and ovarian cancers.

Here, we describe a chemical genetics approach for the identification of PARP-7 substrates based on a previously described NAD$^+$ analog-sensitive PARP (asPARP) approach developed for PARPs 1, 2, and 3 (*Gibson and Kraus, 2017*; *Gibson et al., 2016*). In this asPARP approach, NAD$^+$ with an alkyne-containing R group at position 8 of the adenine ring works in concert with PARP proteins mutated at gatekeeper residues within their active site to facilitate 'click'-able ADP-ribosylation of asPARP-specific protein targets. This approach is conceptually similar to one described by *Carter-O'Connell et al., 2014* in which analog sensitivity is conferred through an addition to the nicotinamide moiety, with click functionality conferred by an alkyne moiety added at position 6 of adenine. We used the asPARP approach described in *Gibson et al., 2016* in combination with mass spectrometry to determine the PARP-7 MARylated proteome, as well as complementary genomic, biochemical, cell-based, and biological analyses. Our studies have revealed an expansive set of PARP-7 protein substrates, including α-tubulin, which reduces microtubule stability when MARylated by PARP-7.

## Results

### PARP-7 regulates cell growth and invasion in ovarian cancer cells

Recent studies linking PARP-7 to cancer (*Cheng et al., 2019*; *Goode et al., 2010*) prompted us to examine the expression of *PARP7* mRNA in human cancer samples. We mined the Genotype-Tissue Expression (GTEx; normal) and The Cancer Genome Atlas (TCGA; cancer) databases to quantify *PARP7* mRNA expression across different normal and cancerous human tissues, including ovary, breast, pancreas, and kidney. Of these four tissues, *PARP7* mRNA was expressed to the highest levels in ovary and the lowest levels in the pancreas (*Figure 1A*, *Figure 1—figure supplement 1A*). The levels of *PARP7* mRNA in cancers compared to the cognate noncancerous tissues varied; ovarian cancer tissues had decreased levels of *PARP7* mRNA, while pancreatic cancers had elevated levels of *PARP7* mRNA, when compared to noncancerous tissues (*Figure 1B*). However, analysis at a more granular level using single-cell RNA-seq data (*Izar et al., 2020*; *Wagner et al., 2020*) revealed that (1) the cell types in normal ovarian tissue are different than those in cancerous tissue and (2) malignant cells from ovarian cancer have a higher average *PARP7* expression level than any of the normal ovarian cell types (*Figure 1—figure supplement 1B*). The latter is supported by data from the Pan-Cancer Atlas in the TCGA database (*Hoadley et al., 2018*) showing a high frequency of *PARP7* gene gains and amplifications (*Figure 1—figure supplement 1C*). Given the difference in cell types and the possibility that ovarian cancers actually arise from cells in the Fallopian tubes (*Erickson et al., 2013*; *Labidi-Galy et al., 2017*), direct comparisons between *PARP7* expression levels in normal and cancerous ovarian tissues are difficult. But, the available data suggest the possibility that gene amplifications drive elevated *PARP7* expression in malignant cells in ovarian cancers, which may portend a dependence of ovarian cancers on PARP-7.

PARP-7 plays multiple essential roles in biological processes, such as transcription, RNA metabolism, translation in response to viral infections, and AHR activation (*Atasheva et al., 2014*; *Bindesbøll et al., 2016*; *Kozaki et al., 2017*; *MacPherson et al., 2013*). To determine the biological role of PARP-7 in ovarian cancers, we performed cell growth, migration, and invasion assays

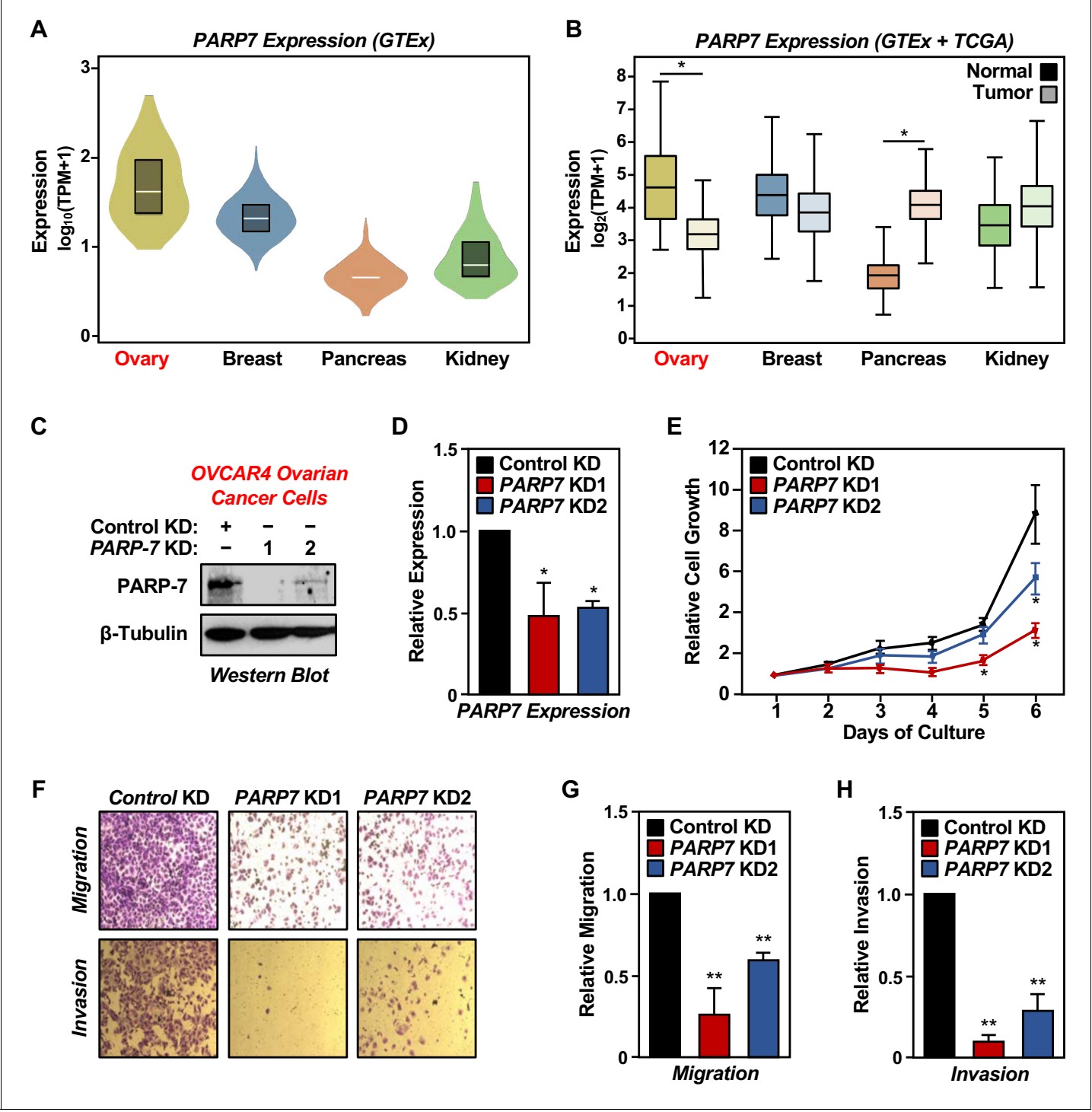

**Figure 1.** PARP-7 expression in cancers and role in ovarian cancer cell phenotypes. (A) Violin plots showing normalized expression of *PARP7* mRNA from GTEx data presented as log₁₀(TPM+1) (Transcripts Per Million), calculated from a gene model with all isoforms collapsed to a single gene. The data were obtained from the GTEx portal and expressed as normalized TPM scores as described (*Li et al., 2010*). Expression profiles of *PARP7* mRNA in the following cancers: breast (n = 459, median TPM = 21.1), ovary (n = 180, median TPM = 37.6), pancreas (n = 328, median TPM = 4.1), and kidney cortex (n = 85, median TPM = 5.6). (B) Box plots showing normalized expression of *PARP7* mRNA across four tissues (ovary, breast, pancreas, and kidney) and their corresponding cancer types (matched TCGA normal and GTEx) analyzed using GEPIA and presented as log₂(TPM+1). Ovary and pancreas show significantly different expression levels for *PARP7* between normal and tumor tissues. Asterisks indicate significant differences between normal and tumor samples (Student's t-test, *p<0.05). (C) Western blots showing PARP-7 protein levels with or without siRNA-mediated knockdown (KD) of *PARP7* in OVCAR4 human ovarian cancer cells. Two different siRNAs were used. β-tubulin was used as loading control. (D) Expression of *PARP7*

*Figure 1 continued on next page*

*Figure 1 continued*

mRNA with or without siRNA-mediated knockdown (KD) of *PARP7* in OVCAR4 cells as determined by RT-qPCR. Each bar represents the mean ± SEM, n = 3. Asterisks indicate significant differences from the control (Student's t-test; *p<0.05). (E) Growth of OVCAR4 cells with or without siRNA-mediated knockdown (KD) of *PARP7* was assayed by crystal violet staining. Each point represents the mean ± SEM, n = 3. Asterisks indicate significant differences from the corresponding control (Student's t-test; *p<0.05). (F) Cell migration assays (*top*) and cell invasion assays (*bottom*) for OVCAR4 cells with or without siRNA-mediated knockdown (KD) of *PARP7*. (G) Quantification of cell migration assays like those shown in (F), *top*. Each bar represents the mean ± SEM, n = 3. Asterisks indicate significant differences from the control (Student's t-test; **p<0.01). (H) Quantification of cell invasion assays like those shown in (F), *bottom*. Each bar represents the mean ± SEM, n = 3. Asterisks indicate significant differences from the control KD (Student's t-test; **p<0.01).

The online version of this article includes the following figure supplement(s) for figure 1:

**Figure supplement 1.** Expression of *PARP7* mRNA in normal and cancerous tissues, and alterations in the *PARP7* gene across four cancer types.
**Figure supplement 2.** Effects of *PARP7* knockdown on cancer-related phenotypes in HeLa, OVCAR3, and A704 cells.

following PARP-7 depletion. siRNA-mediated knockdown of *PARP7*, as determined by western blotting and RT-qPCR (*Figure 1C and D*), resulted in a reduction of OVCAR4 cell growth, migration, and invasion (*Figure 1E-H*). Similar results were observed in three other cell lines: OVCAR3 (ovarian cancer), HeLa (cervical cancer), and A704 (kidney cancer) whose PARP-7 levels are comparable to OVCAR4 cells. In all three cell lines and all three assays, *PARP7* knockdown caused an inhibition of the cancer-related phenotypes (*Figure 1—figure supplement 2*), similar to what we observed in OVCAR4 cells. Thus, the observed effects of PARP-7 on cancer-related endpoints are not restricted to OVCAR4 cells.

## Depletion of PARP-7 alters gene expression in ovarian cancer cells

To determine how PARP-7 might regulate ovarian cancer phenotypes, such as cell growth and migration, we performed RNA-sequencing (RNA-seq) on OVCAR4 human ovarian cancer cells subjected to siRNA-mediated *PARP7* knockdown (*Figure 2A*). We used the gene expression patterns determined by RNA-seq as an indicator of the biological state of the cells. The RNA-seq analysis revealed statistically significant changes in 834 genes, with both increased and decreased expression observed (*Figure 2B* and *Supplementary file 1*). Gene ontology (GO) analyses revealed the enrichment of genes encoding proteins with roles in cell-cell adhesion, cell cycle arrest, apoptosis, and gene regulation (*Figure 2C*). The results from the RNA-seq analysis are consistent with previously described roles of PARP-7 in viral responses and transcription (*Bindesbøll et al., 2016*; *Kozaki et al., 2017*; *MacPherson et al., 2013*), while identifying additional roles for PARP-7 in cell cycle regulation and apoptosis. Multiple GO terms associated with reduced PARP-7 levels, such as cell cycle arrest and cell adhesion, are consistent with observed decreases in cell growth and motility (*Figure 1*).

## Development of an NAD⁺ analog-sensitive PARP-7 (asPARP-7) approach

To understand how loss of PARP-7-mediated MARylation results in the altered cellular state and gene expression found following PARP-7 knockdown, we sought to identify the direct protein substrates of PARP-7. For these studies, we focused on cytoplasmic substrates because PARP-7 is predominantly localized in the cytoplasm of OVCAR4 cells (*Figure 3A*). To identify proteins MARylated by PARP-7, we re-engineered a chemical genetics ('bump-hole', 'analog-sensitive') approach that we previously developed for nuclear PARPs (PARPs 1, 2, and 3) (*Gibson and Kraus, 2017*; *Gibson et al., 2016*) to work with the active site of PARP-7.

In the asPARP approach, a 'gatekeeper' amino acid in the $NAD^+$ binding pocket of a PARP protein is mutated to a smaller residue to create a void ('hole'), which typically reduces the affinity of the PARP for $NAD^+$ and, hence, its catalytic activity. Catalytic activity can be restored, however, by adding a bulky moiety onto $NAD^+$ ('bump'), which fills the void and restores binding (*Figure 3B*). In our asPARP-7 approach, we focused on two $NAD^+$ analog-sensitive gatekeeper residues that we previously identified in PARP-1 (*Gibson and Kraus, 2017*; *Gibson et al., 2016*). Multiple sequence alignment (*Figure 3C*) and structural analysis (*Figure 3D*) of the PARP-1 and PARP-7 ADP-ribosyltransferase domains revealed Phe547 and Ser563 in PARP-7 as homologous to the 1° and 2° gatekeeper mutations previously identified in PARP-1. We generated a series of ten single or double PARP-7 mutants at these two residues, expressed them in insect cells, and purified them along with

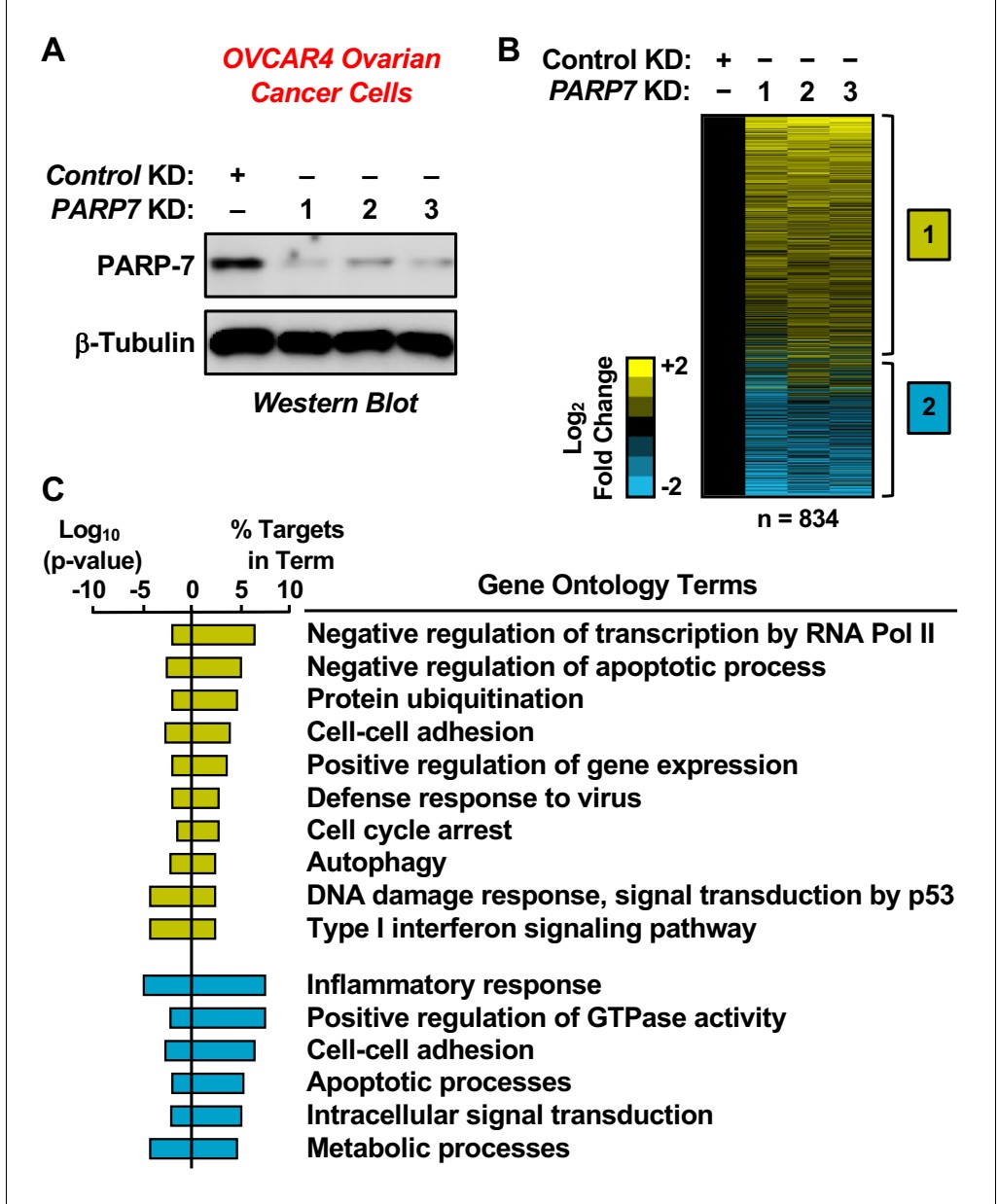

**Figure 2.** RNA-seq analysis of gene expression in ovarian cancer cells following PARP-7 depletion. (A) Western blots showing PARP-7 protein levels after siRNA-mediated knockdown (KD) of *PARP7* in OVCAR4 cells. Three different siRNAs were used. β-tubulin was used as loading control. (B) Heat maps showing the results of RNA-seq assays from OVCAR4 cells upon siRNA-mediated knockdown (KD) of *PARP7*. Three different siRNAs were used. Results represent fold changes in FPKM values for genes significantly regulated versus the control KD, expressed as $\log_2$ fold change. A fold change $\geq 1.5$ was classified as upregulated, while a fold change $\leq -0.5$ was classified as downregulated. (C) Gene ontology terms enriched for the significantly upregulated genes (*yellow*) and significantly downregulated genes (*blue*) shown in (B). The percent of targets from each GO term identified in the analysis is shown, along with the $\log_{10}$ p-value for the enrichment.

wild-type PARP-7 (WT) (*Figure 3E*). We then tested the automodification activity of wild-type and mutant PARP-7 proteins in vitro by incubating purified protein and NAD$^+$, followed by western blotting with a MAR detection reagent (*Gibson et al., 2017*). As expected, wild-type PARP-7 exhibited robust activity, whereas all the mutants exhibited reduced activity, with some showing little activity (*Figure 3F and G*).

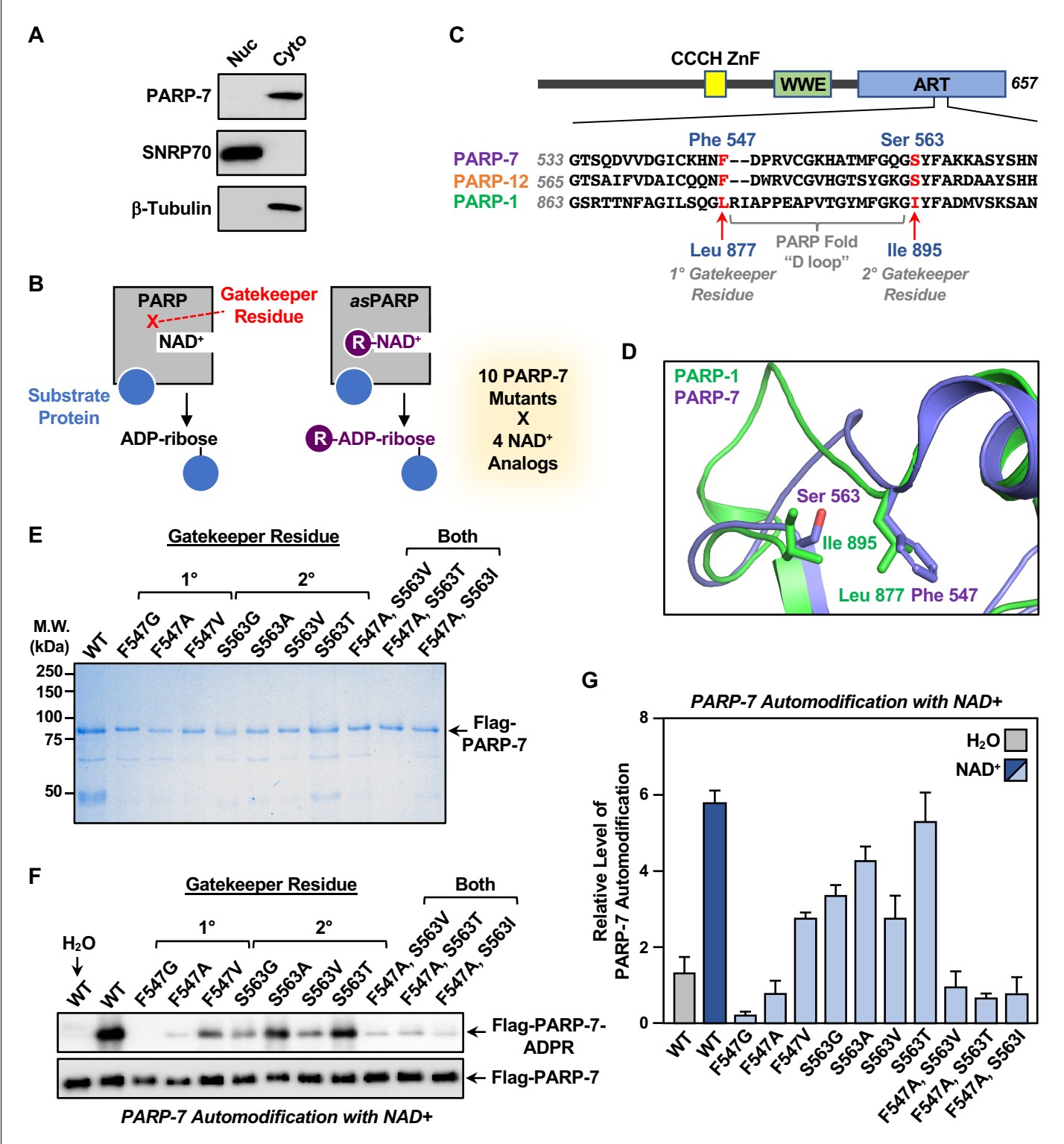

**Figure 3.** Generation of NAD$^+$ analog-sensitive PARP-7 (asPARP-7) mutants. (**A**) PARP-7 localizes to the cytosolic compartment in OVCAR4 cells. Western blots of nuclear and cytosolic fractions of OVCAR4 ovarian cancer cells. SNRP70 and β-tubulin were used as loading controls for the nuclear and cytoplasmic fractions, respectively. (**B**) Schematic diagram illustrating the NAD$^+$ analog-sensitive PARP approach. R, unnatural chemical moieties added to NAD$^+$. In this study, 10 PARP-7 mutants targeting two potential gatekeeper residues (shown in red) were screened with 14 different NAD$^+$ analogs. (**C**) (*Top*) Schematic diagram of PARP-7 showing the functional domains, including the Cys$_3$His$_1$ zinc finger (CCCH ZnF), the WWE PAR-binding domain (WWE), and the ADP-ribosyl transferase domain (ART). (*Bottom*) Multiple sequence alignment of PARP-7 (Q7Z3E1), PARP-1 (P09874) and PARP-

*Figure 3 continued on next page*

*Figure 3 continued*

12 (Q9H0J9). The two potential gatekeeper residues in PARP-7 selected for mutation are highly conserved among PARP-7, PARP-1, and PARP-12. Residues F547 and S563 in PARP-7, as well as the corresponding residues in PARP-1 (residues L877 and I895) and PARP-12 (residues F579 and S595), are indicated in red. (D) Close-up view of residues F547 and S563 in PARP-7. The structure was generated by comparison of available structures of PARP-1 (PDB: 3PAX) and PARP-12 (PDB: 2PQF). The molecular graphic was generated with Pymol. (E) Recombinant PARP-7 proteins used for asPARP-7 activity screening. SDS-PAGE analysis with subsequent Coomassie blue staining of purified FLAG-tagged wild-type (WT) PARP-7 and 10 PARP-7 site-specific mutants expressed in Sf9 insect cells. Molecular weight markers in kilodaltons (kDa) are shown. (F) Western blots showing the automodification (MARylation) activity of the PARP-7 mutants with $NAD^+$ detected using an MAR-binding reagent. PARP-7 was used as loading control for the reactions. (G) Quantification of the automodification activity of the PARP-7 mutants shown in (F).

Next, we tested the catalytic activity of the wild-type and mutant PARP-7 with a set of four $NAD^+$ analogs suitable for copper-catalyzed azide-alkyne cycloaddition ('click') reactions (*Haldón et al., 2015*; *Lutz and Zarafshani, 2008*). Each $NAD^+$ analog has different R groups at position 8 of the adenine ring of $NAD^+$. 8-BuT-6-Parg-$NAD^+$ also contains a modification at position 6 of the adenine ring. Two of the analogs have alkyne-containing R groups (analogs 1 and 2) and two have azide-containing R groups (analogs 3 and 4) (*Figure 4A* and *Supplementary file 2*) to facilitate subsequent click chemistry for purification and fluorescent labeling (*Gibson and Kraus, 2017*; *Gibson et al., 2016*) of MARylated proteins. We assayed wild-type PARP-7 and the ten gatekeeper mutants for $NAD^+$ analog sensitivity in an automodification reaction with water, $NAD^+$, and the four $NAD^+$ analogs (*Figure 4B*). The different PARP-7 mutants exhibited a range of basal activities (water), which likely results from ADP-ribosylation during recombinant protein expression and purification. Automodification activity (i.e. normalized activity) was determined by subtraction of this basal activity for each mutant from the ADP-ribosylation signals observed upon incubation with $NAD^+$ or the four $NAD^+$ analogs (*Figure 4C*).

Three PARP-7 mutants, F547V, S563G and S563T, exhibited some activity with the $NAD^+$ analogs (*Figure 4B and C*). Comparison of the activity of the PARP-7 mutants with natural $NAD^+$ (*Figure 3F and G*) and the four $NAD^+$ analogs (*Figure 4B and C*) revealed that the mutants exhibiting greater activity with $NAD^+$ also showed greater activity with $NAD^+$ analogs. Although this screen yielded three PARP-7 mutants with $NAD^+$ analog sensitivity, we chose to focus on the S563G mutant with 8-Bu(3-yne)T-$NAD^+$ for our subsequent studies. PARP-7 S563G, referred to from here forward as *as*PARP-7, had the highest activity with 8-Bu(3-yne)T-$NAD^+$, a well-characterized and readily available 'click'-able $NAD^+$ analog (*Gibson and Kraus, 2017*; *Gibson et al., 2016*), as seen in an in-gel fluorescence-based assay for 'click'-able autoMARylation (*Figure 5*).

## Identification of PARP-7-specific substrates in ovarian cancer cells using an asPARP-7 approach

Extract-based PARP-specific substrate identification using the asPARP approach relies on poor usage of 8-Bu(3-yne)T-$NAD^+$ by endogenous PARPs compared to the *as*PARP protein. To determine whether addition of *as*PARP-7 to a complex cellular extract results in substantive incorporation of 'click'-able ADP-ribosylation, we incubated OVCAR4 cytoplasmic extract with 8-Bu(3-yne)T-$NAD^+$ and purified recombinant wild-type or *as*PARP-7 protein. 8-Bu(3-yne) T-ADP-ribose-labeled proteins were 'clicked' to an azido-TAMRA fluorophore and imaged using in-gel fluorescence (*Figure 6A*). We found that addition of *as*PARP-7, but not wild-type PARP-7, resulted in strong and specific incorporation of 'click'-able ADP-ribose onto OVCAR4 cytoplasmic extract proteins.

To identify PARP-7 target proteins by mass spectrometry, we incubated cytoplasmic extracts from HeLa or OVCAR4 cells with 8-Bu(3-yne)T-$NAD^+$ and *as*PARP-7, and then clicked the 8-Bu(3-yne)T-ADP-ribose–labeled proteins to azide-agarose. Following extensive washing of the agarose beads, we performed trypsin-based identification of the ADP-ribosylated proteins by mass spectrometry (*Supplementary files 3* and *4*). We identified ~500 high confidence protein substrates of PARP-7 common to both OVCAR4 and HeLa cells, as well as an additional ~500 high confidence protein substrates of PARP-7 each unique to OVCAR4 and HeLa cells (*Figure 6B*). GO analyses of the common PARP-7 substrates revealed functions in translation, rRNA processing, cytoskeleton organization, and cell proliferation, among others (*Figure 6C*). These results provide important clues to the direct biological effects of PARP-7.

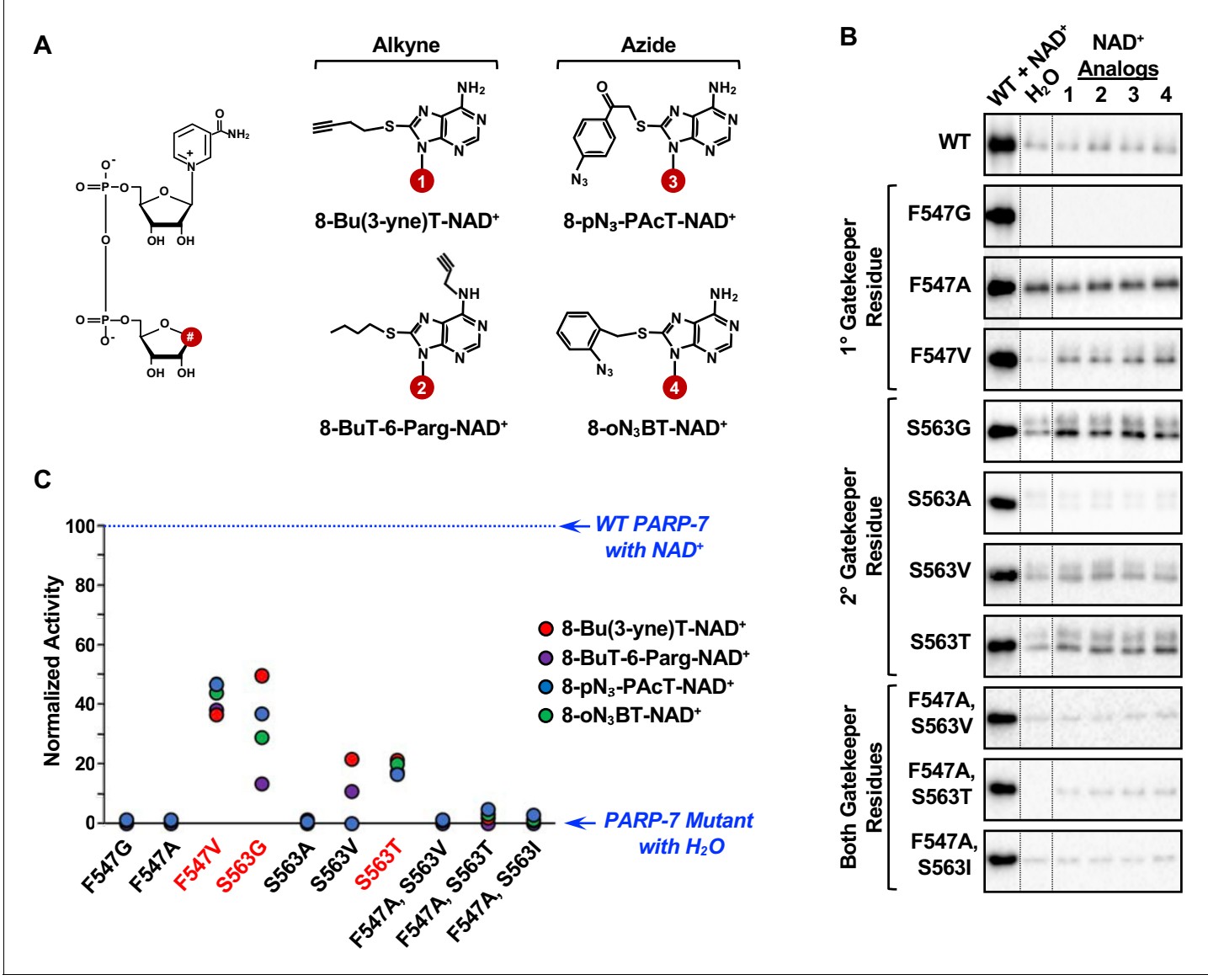

**Figure 4.** Screening the activity of potential analog-sensitive PARP-7 mutants with clickable NAD⁺ analogs. (**A**) Chemical structures of four clickable NAD⁺ analogs used for the asPARP-7 screen. (*Left*) The R groups are linked at position 8 (#) of the adenine ring of NAD⁺. (*Right*) 8-BuT-6-Parg-NAD⁺ and 8-Bu(3-yne)T-NAD⁺ are clickable through their alkyne groups, whereas 8-oN₃-BT-NAD⁺ and 8-pN3-PAcT-NAD⁺ are clickable through their azide groups. (**B**) Western blots showing the wild-type and mutant PARP-7 automodification reactions performed with NAD⁺ or the clickable NAD⁺ analogs shown in (**A**). The MAR signals were detected using a MAR binding reagent. (**C**) Normalized automodification activity of wild-type and mutant PARP-7 proteins with NAD⁺ or the clickable NAD⁺ analogs shown in (**A**). PARP-7 automodification assays like those shown in (**B**) were used to determine the activity of the mutants with the NAD⁺ analogs. This screen was performed twice and yielded similar results each time; the results from one replicate are shown.

## PARP-7 MARylates α-tubulin to regulate microtubule stability in ovarian cancer cells

Our asPARP-7 plus mass spectrometry approach identified α-tubulin as a target for PARP-7-mediated MARylation. Our results are consistent with a recent report indicating that PARP-7 colocalizes with and MARylates α-tubulin in cells (*Grimaldi et al., 2019*). We first confirmed that α-tubulin was MARylated in OVCAR4 cells by immunoprecipitating MARylated proteins and assaying for α-tubulin by western blotting (*Figure 7A*). Next, we immunoprecipitated MARylated proteins from OVCAR4 cells subjected to either control or *PARP7* knockdown and assayed by western blotting for the

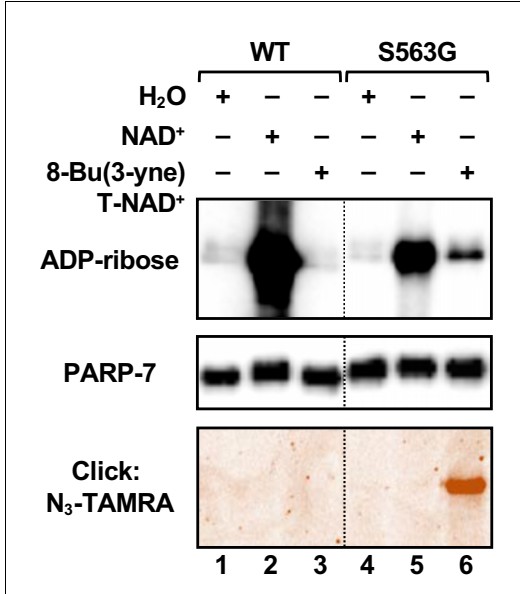

**Figure 5.** Activity of the S563G analog-sensitive PARP-7 mutant with 8-Bu(3-yne)T-NAD$^+$. Automodification reactions with wild-type and S563G mutant PARP-7 performed with NAD$^+$ or 8-Bu(3-yne)T-NAD$^+$. The autoMARylation signals were detected by (*top*) western blotting using a MAR binding reagent or (*bottom*) click chemistry-based in-gel fluorescence. PARP-7, detected by western blotting (*middle*), was used as loading control for the reactions. TAMRA, tetramethylrhodamine.

presence of α-tubulin, confirming that PARP-7 was required for MARylation of α-tubulin (*Figure 7A and B*).

Microtubule architecture, mediated in part by α-tubulin, plays a key role in cell migration and mitotic progression (*Boggs et al., 2015*; *Pillai et al., 2015*; *Piperno et al., 1987*). Given the observed effects of PARP-7 depletion on mRNAs and proteins involved in cell cycle regulation and cytoskeleton organization (*Figures 2C* and *6C*), as well the MARylation of α-tubulin by PARP-7, we surmised that the functional interplay between PARP-7 and α-tubulin might underlie microtubule dynamics. To test this directly, we assayed how OVCAR4 cells with or without *PARP7* knockdown recovered from microtubule depolymerization by performing immunostaining for α-tubulin with visualization by confocal fluorescence microscopy. The cells were first incubated at a cold temperature, which destabilizes the microtubules, and then moved to 37°C to allow the regrowth of microtubules. Depletion of PARP-7 by knockdown of *PARP7* mRNA stabilized the cold-destabilized α-tubulin-containing microtubule structures under these conditions (*Figure 7C*). Similar results were observed in a parallel experiment in cells treated with Nocodazole, a drug that depolymerizes microtubules (*Pillai et al., 2015*; *Figure 7C*). Similar results were observed in three other cell lines: OVCAR3, HeLa, and A704 (*Figure 7—figure supplement 1*), indicating that the effects are not restricted to ovarian cancer cells. The levels of α-tubulin MARylation and PARP-7 increase after treatment with cold or Nocodazole (*Figure 7D*), linking PARP-7 levels, α-tubulin MARylation, and microtubule depolymerization.

To connect the catalytic activity of PARP-7 more directly to the function of α-tubulin in cells, we used two approaches. First, we used a catalytically dead PARP-7 mutant (Y564A), which we expressed using a Flag-tagged mouse *Parp7* cDNA to produce an siRNA-resistant mRNA that could be used in knockdown-addback experiments (the *PARP7* siRNAs that we used were designed against the human *PARP7* mRNA). Ectopic expression of the catalytically dead PARP-7 mutant resulted in a loss of α-tubulin MARylation (*Figure 7—figure supplement 2A*). In addition, expression of the catalytically dead PARP-7 mutant after depletion of endogenous PARP-7 phenocopied *PARP7* knockdown in the microtubule stability assay (*Figure 7—figure supplement 2B*; compare to *Figure 7C*) and the cell migration assay (*Figure 7—figure supplement 2, C and D*; compare to *Figure 1F and G*) (i.e. it was impaired versus wild-type PARP-7 in both cases). These results demonstrate that PARP-7 catalytic activity is required for the observed effects of PARP-7.

Second, we identified the sites of PARP-7-mediated MARylation on α-tubulin using our asPARP-7 approach, mutated them, and performed functional analyses with the MARylation site mutant. To do so, we eluted trypsin-digested, 8-Bu(3-yne)T-ADP-ribose-labeled proteins from the azide-agarose beads described above using hydroxylamine and analyzed them by mass spectrometry. Hydroxylamine-cleaved ADPR modifications produce a 15.019 *m/z* shift identifying the specific site of glutamate or aspartate modification (*Zhang et al., 2013*). Our analysis identified multiple sites of PARP-7-mediated MARylation on α-tubulin (*Supplementary file 5*). We focused on three residues in the critical amino-terminal nucleotide-binding domain (i.e. D69, E71, E77), all of which were contained in a single peptide that was identified in all four replicates (two each for OVCAR4 and HeLa cells). We mutated all three of these residues to similar, but non-modifiable, residues (D → N and E → Q) and then expressed the mutant protein in OVCAR4 cells. Mutation of these sites impaired MARylation of

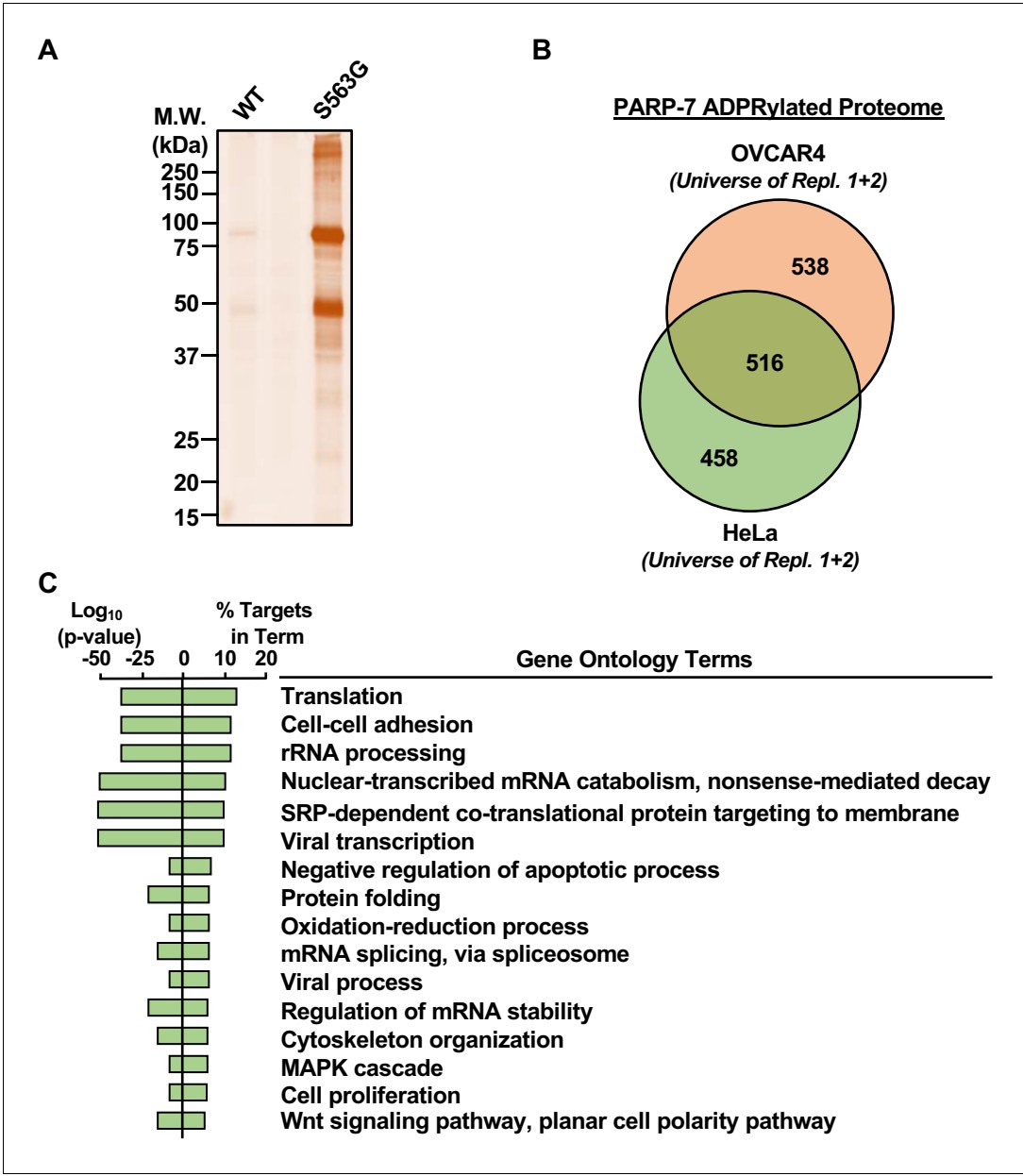

**Figure 6.** Identification of PARP-7-specific substrates using an NAD$^+$ analog-sensitive PARP-7 approach. (**A**) In-gel fluorescence of cytoplasmic extract from OVCAR4 cells conjugated to azido-TAMRA after labeling reactions with 8-Bu(3-yne)T-NAD$^+$ in the presence of wild-type or S563G mutant PARP-7. Molecular weight markers in kDa are shown. Similar assays were performed with cytoplasmic extract from HeLa cells. The assay was performed twice in each cell line. (**B**) Venn diagram depicting the overlap of the protein substrates of PARP-7-mediated MARylation between OVCAR4 and HeLa cells identified using asPARP-7 and mass spectrometry. (**C**) Gene ontology terms enriched for the protein targets of PARP-7-mediated MARylation common between OVCAR4 and HeLa cells.

α-tubulin (*Figure 7E*), and ectopic expression of the α-tubulin mutant phenocopied *PARP7* knock-down and expression of a catalytically dead PARP-7 mutant in the microtubule stability assay (*Figure 7F and G*; compare to *Figure 7C* and *Figure 7—figure supplement 2B*). These results show that direct MARylation of α-tubulin is required for the observed effects of PARP-7.

Collectively, these results indicate that PARP-7 reduces cellular microtubule content following recovery from depolymerization, likely in part through direct MARylation of α-tubulin. Moreover, the PARP-7-mediated microtubule control may play a role in the regulation of cancer cell growth and motility.

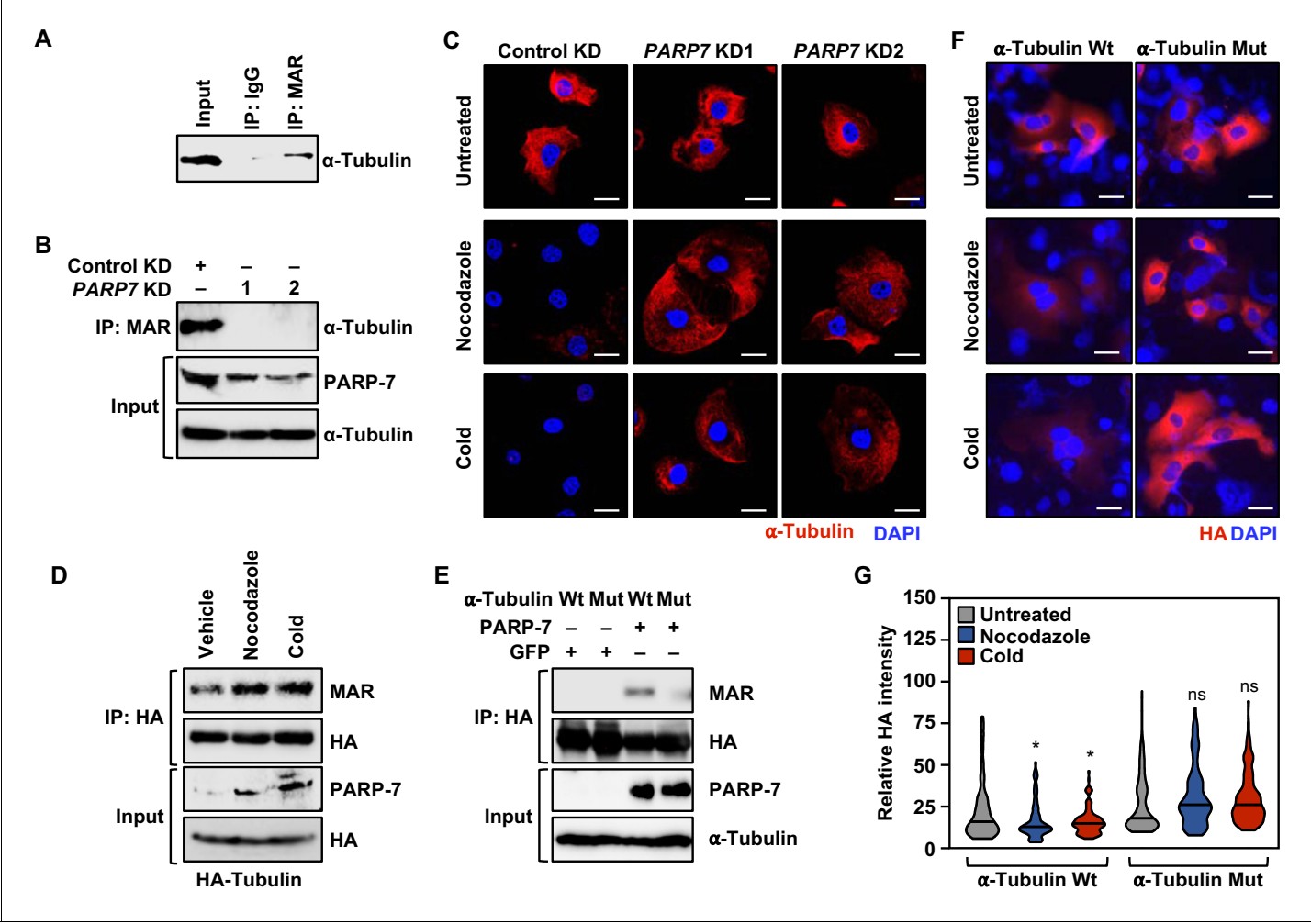

**Figure 7.** PARP-7 links MARylation of α-tubulin to the regulation of microtubule stability in ovarian cancer cells. (**A**) MARylation of α-tubulin in OVCAR4 cells. MARylated proteins were immunoprecipitated from OVCAR4 cells using a MAR detection reagent and subjected to western blotting for α-tubulin. The experiment was performed three times (n = 3) to ensure reproducibility. (**B**) Knockdown of PARP-7 reduces the MARylation of α-tubulin. MARylated proteins were immunoprecipitated from OVCAR4 cells after siRNA-mediated knockdown (KD) of *PARP7* and the subjected to western blotting for α-tubulin. The experiment was performed three times (n = 3) to ensure reproducibility. (**C**) Knockdown of PARP-7 promotes microtubule stability in OVCAR4 cells. Immunofluorescent staining of α-tubulin in OVCAR4 cells after siRNA-mediated knockdown (KD) of *PARP7* and treatment with cold or nocodazole. Scale bars = 25 µm. The experiment was performed three times (n = 3) to ensure reproducibility. (**D**) Cold or nocodazole treatment increases MARylation of α-tubulin. HA-tagged α-tubulin was immunoprecipitated from OVCAR4 cells subjected to cold or nocodazole treatment using an HA antibody. The IP material was subjected to western blotting for MAR and HA, and the input material was subjected to western blotting for PARP-7 and HA. (**E**) Ectopic expression of PARP-7 enhances the MARylation of α-tubulin. Wild-type (Wt) or MAR-deficient mutant α-tubulin was immunoprecipitated from HEK 293 T cells after ectopic expression of PARP-7. The IP material was subjected to western blotting for MAR and HA, and the input material was subjected to western blotting for PARP-7 and α-tubulin. (**F**) Inhibition of α-tubulin MARylation promotes microtubule stability in OVCAR4 cells. Immunofluorescent staining of HA-epitope tagged wild-type or MARylation site mutant α-tubulin in OVCAR4 cells after treatment with cold or nocodazole. Scale bars = 100 µm. The experiment was performed three times (n = 3) to ensure reproducibility. (**G**) Quantification of α-tubulin staining from experiments like those shown in panel (**F**). Asterisks indicate significant differences from the control (n = 3; ANOVA, *p<0.05).

The online version of this article includes the following figure supplement(s) for figure 7:

**Figure supplement 1.** Effects of *PARP7* knockdown on microtubule stability in HeLa, OVCAR3, and A704 cells.

**Figure supplement 2.** PARP-7 catalytic activity is required for MARylation of α-tubulin and functional outcomes.

## Discussion

Studies over the past two decades have begun to elucidate the biological functions of PARP-7. The earliest studies focused on the role of PARP-7 as a component of the AHR signaling pathway in response to the environmental contaminant dioxin (TCDD) (*Ma et al., 2001*; *MacPherson et al., 2013*). This led to the observation that PARP-7 is a negative regulator of AHR activity that acts to reduce sensitivity to dioxin-induced steatohepatitis (*Ahmed et al., 2015*; *Hutin et al., 2018*). More recent studies have implicated PARP-7 in immune responses to viral infections (*Atasheva et al., 2014*; *Kozaki et al., 2017*), maintenance of pluripotency in embryonic stem cells (*Roper et al., 2014*), and promotion of neural progenitor cell proliferation (*Grimaldi et al., 2019*). And, as discussed in more detail below, PARP-7 has also been implicated in cancer, although its role has been less clear and may vary in different cancer types. Herein, we described a chemical genetics approach that has allowed us to identify substrates of PARP-7, providing insights into the biology of PARP-7.

### Substrate identification as a window into the biology of PARP-7

Information on the protein substrates of PARP-7 has been limited, although previous studies have identified a few targets, including PARP-7 itself through automodification, as well as histones, AHR, and LXRs (*Bindesbøll et al., 2016*; *Gomez et al., 2018*; *Ma et al., 2001*; *MacPherson et al., 2013*). To identify the MARylated substrates that underlie PARP-7-mediated ovarian cancer cell phenotypes, we developed a chemical genetics approach for $NAD^+$ analog-sensitivity comprising an $NAD^+$ binding pocket PARP-7 mutant (S563G) paired with the $NAD^+$ analog 8-Bu(3-yne)T-$NAD^+$ (*Figures 3–5*). We used this approach coupled with mass spectrometry to identify an extensive PARP-7 ADP-ribosylated proteome in OVCAR4 and HeLa cells, including cell-cell adhesion and cytoskeleton organization proteins (*Figure 6C*). Interestingly, similar gene ontologies were enriched in our gene expression experiments in OVCAR4 cells after knockdown of *PARP7* (*Figure 2C*), suggesting a coordination of the regulatory functions of PARP-7.

One PARP-7 substrate that we identified, α-tubulin, has links to cancer, playing a role in cell migration and mitotic progression (*Boggs et al., 2015*; *Pillai et al., 2015*; *Piperno et al., 1987*). A previous study demonstrated that MARylation of α-tubulin by PARP-7 supports proper organization of the mouse neuronal cortex, including the correct distribution and number of GABAergic neurons (*Grimaldi et al., 2019*). In our studies, we observed that PARP-7 reduces cellular microtubule content following recovery from depolymerization, likely in part through direct MARylation of α-tubulin (*Figure 7*). This may result from changes in the polymerization rate, tubulin dimer stability, microtubule stability, etc. PARP-7-mediated microtubule control is likely to underlie the regulation of cell growth and motility that we observed in ovarian cancer cells (*Figure 1C-H*).

Why might PARP-7 MARylate α-tubulin to destabilize microtubules in ovarian cancer cells? How does this benefit the cancer cells? One possibility is that the ability to quickly depolymerize and disassemble microtubules is important for efficient cell proliferation and migration. Indeed, there is evidence for this in the literature. Taxanes and related molecules comprise a class of anticancer drugs that stabilize microtubules, leading to the arrest of proliferation and mitosis. Moreover, previous studies have shown that well-characterized tumor suppressors, such as RASSF1A and APC, act to stabilize microtubule polymerization (*Liu et al., 2003*; *van Es et al., 2001*). These observations provide a plausible explanation for the observed effects of PARP-7 on α-tubulin and microtubules in our assays. Rodriguez et al. (co-submitted) did not identify α-tubulin as a PARP-7 substrate in their analysis, but they did identify tubulin-specific chaperone E, a key chaperone for α-tubulin (*Tian and Cowan, 2013*), perhaps indicating a broader role for PARP-7 in regulating microtubule formation.

Together, our results demonstrate a functional link between PARP-7, its catalytic activity toward a specific substrate (i.e. α-tubulin), a defined cellular process (i.e. microtubule control), and a broader biological outcome (i.e. cancer-related phenotypes).

### Interplay among PARP family members

Among the PARP-7 substrates that we identified in our analyses, we found other PARP family members, including PARP-1 (both replicates from OVCAR4 and HeLa cells), PARP-4 (one replicate from OVCAR4 cells and both replicates from HeLa cells), and PARP-2 (one replicate from HeLa cells) (*Supplementary file 3*). PARPs 1 and 2 are nuclear proteins with functions in DNA repair and transcription (*Gupte et al., 2017*; *Ryu et al., 2015*), whereas PARP-4 (a.k.a. vault PARP) is a component

of enigmatic cell structures known as vault particles (*Kickhoefer et al., 1999*). In previous studies of PARPs 1, 2, and 3, we observed extensive crosstalk between them, each acting as a substrate for the other (*Gibson et al., 2016*). Such crosstalk may be a universal feature of PARPs. In this regard, Rodriguez et al. (co-submitted) identified PARP-13 as a substrate of PARP-7 using a similar chemical genetics approach, which also included MARylation site identifications. Their analysis identified cysteine as a major ADPR acceptor for PARP-7, consistent with observations by Gomez et al. implicating cysteines and acidic residues as targets for MARylation (*Gomez et al., 2018*).

## Context-specific roles of PARP-7 in cancers and therapeutic opportunities

*TIPARP* (the gene encoding PARP-7, located at 3q25) was identified in a susceptibility locus for ovarian cancer in a recent genome-wide association study (*Goode et al., 2010*). We have recently shown that ADP-ribosylation levels and patterns correlate with gene expression and clinical outcomes in ovarian cancers (*Conrad et al., 2020*). In addition, we and others have observed that *PARP7* mRNA expression is lower in ovarian cancers compared to normal ovarian epithelium, both in patient samples (*Figure 1B*) and cell lines (*Goode et al., 2010*). However, a more detailed analysis using single cell RNA-seq data (*Izar et al., 2020*; *Wagner et al., 2020*) revealed that malignant cells from ovarian cancer have a higher average *PARP7* expression level than any of the normal ovarian cell types (*Figure 1—figure supplement 1B*). The latter is supported by data from the Pan-Cancer Atlas in the TCGA database (*Hoadley et al., 2018*) showing a high frequency of *PARP7* gene gains and amplifications (*Figure 1—figure supplement 1C*). Although direct comparisons between *PARP7* expression level in normal and cancerous ovarian tissues are difficult, the available data suggest the possibility that gene amplifications drive elevated *PARP7* expression in malignant cells in ovarian cancers.

In breast cancers, *PARP7* mRNA expression is lower in tumor tissues compared to normal tissues, and higher *PARP7* mRNA is associated with better survival outcomes (*Cheng et al., 2019*; *Zhang et al., 2020*). Moreover, in breast and colon cancer cells, PARP-7 acts to suppress multiple oncogenic transcription factors, including HIF-1α, to reduce tumorigenesis (*Zhang et al., 2020*). Knockdown of *PARP7* in breast and colon cancer xenografts promotes enhanced tumor formation (*Zhang et al., 2020*).

These observations stand in contrast to our results with ovarian cancer cells, in which *PARP7* knockdown resulted in reduced cell growth, migration, and invasion (*Figure 1C-H*), as well as increased microtubule content, which may reduce cancer-related outcomes. Thus, PARP-7 may have context-specific effects in cancer cells. In ovarian cancers and other cancers that respond similarly to *PARP7* knockdown, inhibition of PARP-7 catalytic activity with small molecules would be expected to have positive therapeutic effects. In this regard, the first PARP-7 inhibitor (RBN-2397) is now in Phase I clinical trials for solid tumors (ClinicalTrials.gov identifier: NCT04053673). In contrast to the current FDA-approved PARP1/2 inhibitors, RBN-2397 is the first inhibitor of a MART, representing a previously unexplored therapeutic target.

Finally, treatment with carboplatin/paclitaxel is the primary initial treatment regimen for the vast majority of ovarian cancers (*Boyd and Muggia, 2018*; *Kampan et al., 2015*). Interestingly, paclitaxel is a microtubule-stabilizing drug that prevents mitosis (*Orr et al., 2003*; *Weaver, 2014*). Although paclitaxel binds specifically to β-tubulin (*Orr et al., 2003*), overexpression of α-tubulin can increase the resistance of cancer cells to paclitaxel (*Han et al., 2000*). These observations, viewed in light of our results, suggest an intriguing link between PARP-7, α-tubulin, microtubules, and the sensitivity of cancer cells to paclitaxel.

## Materials and methods

### Key resources table

| Reagent type (species) or resource | Designation | Source or reference | Identifiers | Additional information |
|---|---|---|---|---|
| Gene (*Homo sapiens*) | TIPARP (PARP7) | Genbank | Gene ID: 25976 | |

*Continued on next page*

*Continued*

| Reagent type (species) or resource | Designation | Source or reference | Identifiers | Additional information |
|---|---|---|---|---|
| Gene (*Mus musculus*) | Tiparp (Parp7) | Genbank | GeneID: 99929 | |
| Gene (*Homo sapiens*) | TUBA1A | Genbank | GeneID: 7846 | |
| Strain, strain background (*Escherichia coli*) | DH10BAC | ThermoFisher | Cat#10361012 | |
| Cell line (*Homo sapiens*) | OVCAR4 (female adult ovarian cancer) | ATCC | RRID:CVCL_1627 | |
| Cell line (*Homo sapiens*) | OVCAR3 (female adult ovarian cancer) | ATCC | RRID:CVCL_0465 | |
| Cell line (*Homo sapiens*) | HeLa (female adult cervical cancer) | ATCC | RRID:CVCL_0030 | |
| Cell line (*Homo sapiens*) | A704 (male adult kidney cancer) | ATCC | RRID:CVCL_1065 | Maintained and obtained from Dr. Laura Banaszynski, UT Southwestern |
| Cell line (*Homo sapiens*) | HEK 293T (normal embryonic kidney) | ATCC | RRID:CVCL_0063 | |
| Cell line (*Spodoptera frugiperda*) | Sf9 | ATCC | RRID:CVCL_0549 | |
| Transfected construct (*Homo sapiens*) | *PARP7* siRNA #1 | Sigma-Aldrich | Cat# D-013948–08 | |
| Transfected construct (*Homo sapiens*) | *PARP7* siRNA #2 | Sigma-Aldrich | Cat# D-013948–09 | |
| Transfected construct (*Homo sapiens*) | *PARP7* siRNA #4 | Sigma-Aldrich | Cat# D-013948–07 | Our *PARP7* siRNA #3 |
| Antibody | Anti-mono-ADP-ribose binding reagent (rabbit monoclonal; IgG Fc) | Millipore, MABE1076 | RRID:AB_2665469 | *Gibson and Kraus, 2017* WB (8 µg/mL) IP (3 µg) |
| Antibody | PARP-7 (rabbit polyclonal) | Invitrogen, PA5-40774 | RRID:AB_2607074 | WB (1:500) |
| Antibody | α-tubulin (mouse monoclonal) | Santa Cruz, sc-8035 | RRID:AB_628408 | WB (1:1000) IF (1:1000) |
| Antibody | β-tubulin (rabbit polyclonal) | Abcam, ab6046 | RRID:AB_2210370 | WB (1:1000) |
| Antibody | SNRP70 (rabbit polyclonal) | Abcam, ab51266 | RRID:AB_10673827 | WB (1:1000) |
| Antibody | Flag (mouse monoclonal) | Sigma-Aldrich, F3165 | RRID:AB_259529 | WB (1:2000) |
| Antibody | HA (mouse monoclonal) | Sigma-Aldrich, H3663 | RRID:AB_262051 | WB (1:1000) IF (1:200) IP (1.5 µg) |

*Continued on next page*

Continued

| Reagent type (species) or resource | Designation | Source or reference | Identifiers | Additional information |
|---|---|---|---|---|
| Antibody | Rabbit IgG fraction (polyclonal) | Invitrogen, 10500C | RRID:AB_2532981 | IP (3 μg) |
| Antibody | HRP-conjugated anti-rabbit IgG (goat polyclonal) | Pierce, 31460 | RRID:AB_228341 | WB (1:5000) |
| Antibody | HRP-conjugated anti-mouse IgG (goat polyclonal) | Pierce, 31430 | RRID:AB_228307 | WB (1:5000) |
| Antibody | Alexa Fluor 488-conjugated anti-mouse IgG (goat polyclonal) | ThermoFisher, A-11001 | RRID:AB_2534069 | IF (1:500) |
| Recombinant DNA reagent | pFastBac1 | ThermoFisher | Cat# 10360014 | |
| Recombinant DNA reagent | pFastBac1-human wild-type PARP-7 | This study | | |
| Recombinant DNA reagent | pFastBac1-human analog sensitive PARP-7 | This study | | |
| Recombinant DNA reagent | pcDNA3.1 | ThermoFisher | Cat# V79020 | |
| Recombinant DNA reagent | pcDNA3.1-mouse wild-type PARP-7 | This study | | |
| Recombinant DNA reagent | pcDNA3.1-mouse catalytic mutant PARP-7 | This study | | |
| Recombinant DNA reagent | pINDUCER20 | Addgene | Plasmid no. 44012 | |
| Recombinant DNA reagent | pINDUCER20-mouse wild-type PARP-7 | This study | | |
| Recombinant DNA reagent | pINDUCER20-mouse catalytic mutant PARP-7 | This study | | |
| Recombinant DNA reagent | pCMV-VSV-G | Addgene | Plasmid no. 8454 | |
| Recombinant DNA reagent | pAd-VAntage | Promega | Cat# TB207 | |
| Recombinant DNA reagent | psPAX2 | Addgene | Plasmid no. 12260 | |
| Recombinant DNA reagent | pCMV3-C-HA-human wild-type-α-tubulin | Sino Biologicals | Cat# HG14201-CY | |
| Recombinant DNA reagent | pCMV3-C-HA-human MARylation site mutant-α-tubulin | This study | | |
| Commercial assay or kit | Bac-to-Bac Baculovirus Expression System | ThermoFisher | Cat# 10359016 | |

*Continued*

| Reagent type (species) or resource | Designation | Source or reference | Identifiers | Additional information |
|---|---|---|---|---|
| Commercial assay or kit | RNAeasy Plus Mini kit | Qiagen | Cat# 74136 | |
| Chemical compound, drug | 8-Bu (3-yne)T-NAD$^+$ | BIOLOG Life Science Institute | Cat# N 055 | |
| Software, algorithm | FastQC | Babraham Bioinformatics | http://www.bioinformatics.babraham.ac.uk/projects/fastqc/ | |
| Software, algorithm | Tophat | *Trapnell et al., 2010* | https://ccb.jhu.edu/software/tophat/index.shtml | |
| Software, algorithm | Cufflinks | *Trapnell et al., 2010*; *Trapnell et al., 2012* | http://cole-trapnell-lab.github.io/cufflinks/ | |
| Software, algorithm | DAVID Bioinform-atics Resources | LHRI *Huang et al., 2009a*; *Huang et al., 2009b* | https://david.ncifcrf.gov/home.jsp | |
| Software, algorithm | Java TreeView | *Saldanha, 2004* | http://jtreeview.sourceforge.net/ | |
| Software, algorithm | GTEx Analysis Release v. 8 | The Genotype-Tissue Expression (GTEx) project | https://gtexportal.org/home/ | |

## Antibodies

The custom recombinant antibody-like MAR binding reagent (anti-MAR) was generated and purified in-house (now available from Millipore Sigma, MABE1076; RRID:AB_2665469) (*Gibson et al., 2017*). The antibodies used were as follows: PARP-7 (Invitrogen, PA5-40774; RRID:AB_2607074), α-tubulin (Santa Cruz, sc-8035; RRID:AB_628408), β-tubulin (Abcam, ab6046; RRID:AB_2210370), SNRP70 (Abcam, ab51266; RRID:AB_10673827), Flag (Sigma-Aldrich, F3165; RRID:AB_259529), HA (Sigma-Aldrich, H3663; RRID:AB_262051), rabbit IgG fraction (Invitrogen, 10500C; RRID:AB_2532981), goat anti-rabbit HRP-conjugated IgG (Pierce, 31460; RRID:AB_228341), goat anti-mouse HRP-conjugated IgG (Pierce, 31430; RRID:AB_228307), and Alexa Fluor 488 goat anti-mouse IgG (ThermoFisher, A-11001; RRID:AB_2534069).

## Mining public databases of human samples

### Determining tissue expression using the GTEx data portal

The Genotype-Tissue Expression (GTEx) (https://gtexportal.org/home/) project is a resource which allows users to study human gene expression and its relationship to genetic variation. We used GTEx Analysis Release ver. 8 (dbGaP Accession phs000424.v8.p2) to determine the expression of *PARP7* in normal human tissues. We selected four tissue types: ovary (n = 180), breast (n = 459), pancreas (n = 328), and kidney cortex (n = 85). The expression values for each tissue, calculated in TPM (Transcripts Per Million, determined using a gene model with isoforms collapsed to a single genes), were downloaded from the GTEx portal and presented in violin plots showing expression values in $\log_{10}(TPM+1)$.

### Analysis of expression data in tumor and normal samples from TCGA using GEPIA

GEPIA is a portal for analyzing gene expression with a standardized pipeline using RNA-sequencing data from 9736 tumor and 8587 normal samples obtained from TCGA and GTEx (*Tang et al., 2019*). We compared expression levels of *PARP7* in four different cancer types: ovary, breast, pancreas, and

kidney. We used the GEPIA portal (http://gepia2.cancer-pku.cn/#analysis) to generate box plots for comparing expression in the four cancer types across TCGA matched to normal samples from GTEx. The patients in each cohort group were as follows: ovary (tumor = 426, normal = 88), breast (tumor = 1085, normal = 291), pancreas (tumor = 179, normal = 171), kidney (tumor = 523, normal = 100). The expression data were $\log_2$(TPM+1) transformed for differential analysis, which is plotted in our analysis.

## Analysis of somatic mutations, copy number alterations, and expression using TCGA data sets

Ovarian cancer mutation, alteration, and expression data were accessed from Pan-Cancer Atlas (*Hoadley et al., 2018*; https://portal.gdc.cancer.gov/legacy-archive/search/f) in the Cancer Genome Atlas (TCGA) (https://confluence.broadinstitute.org/display/GDAC/Home). For mRNA expression analysis in relation to somatic mutations and copy number alterations, bar graphs from the ovarian cancer cohort of TCGA were downloaded from cBioPortal (http://www.cbioportal.org/index.do) (*Cerami et al., 2012*; *Gao et al., 2013*).

## Mammalian cell culture

OVCAR4, OVCAR3, 293T, and HeLa cells were purchased from the American Type Cell Culture (ATCC). A704 cells were obtained from Dr. Laura Banaszynski. OVCAR4 and OVCAR3 cells were maintained in RPMI (Sigma-Aldrich, R8758) supplemented with 10% fetal bovine serum and 1% penicillin/streptomycin. HeLa, 293T and A704 cells were maintained in DMEM (Sigma-Aldrich, D5796) supplemented with 10% fetal bovine serum and 1% penicillin/streptomycin. Fresh cell stocks were regularly replenished from the original ATCC-verified stocks and confirmed as mycoplasma-free every three months using a commercial testing kit.

## siRNA-mediated knockdown

The siRNAs targeting PARP-7 were from Dharmacon (see below) and the control siRNA (SIC001) was from Sigma. The siRNA oligos were transfected into OVCAR4 cells at a final concentration of 30 nM using Lipofectamine RNAiMAX reagent (Invitrogen, 13778150) according to the manufacturer's instructions. The cells were used for various assays 24 hr after siRNA transfection. The siRNA oligos used to knockdown *PARP7* mRNA were as follows:

- PARP7 siRNA 1: D-013948–08 (our siRNA #1)
- PARP7 siRNA 2: D-013948–09 (our siRNA #2)
- PARP7 siRNA 4: D-013948–07 (our siRNA #3)

## Preparation of cell lysates and western blotting

OVCAR4 cells were cultured as described above before the preparation of whole cell lysates, nuclear extracts, and cytoplasmic extracts.

### Preparation of whole cell lysates

The cells were collected, washed twice with ice-cold PBS, and resuspended in Lysis Buffer (20 mM Tris-HCl pH 7.5, 150 mM NaCl, 1 mM EDTA, 1 mM EGTA, 1% NP-40, 1% sodium deoxycholate, 0.1% SDS) containing 1 mM DTT and 1x complete protease inhibitor cocktail (Roche, 11697498001). The cells were vortexed for 30 s in Lysis Buffer and then centrifuged to remove the cell debris. The supernatants were collected, aliquoted, flash frozen in liquid nitrogen, and stored at −80°C until used.

### Preparation of nuclear and cytoplasmic extracts

The cells were collected, washed twice with ice-cold PBS, and resuspended in Isotonic Buffer (10 mM Tris-HCl pH 7.5, 2 mM MgCl$_2$, 3 mM CaCl$_2$, 0.3 M sucrose, with freshly added 1 mM DTT, and 1x protease inhibitor cocktail, 250 nM ADP-HPD, and 10 μM PJ34), incubated on ice for 15 min, and lysed by the addition of 0.6% IGEPAL CA-630 with gentle vortexing. After centrifugation, the nuclei were collected by centrifugation in a microcentrifuge for 1 min at 4°C at 1000 rpm. The supernatant was collected as the cytoplasmic fraction. The pelleted nuclei were resuspended in Nuclear

Extraction Buffer (20 mM HEPES pH 7.9, 1.5 mM MgCl$_2$, 0.42 M NaCl, 0.2 mM EDTA, 25% v/v glycerol, with freshly added 1 mM DTT, 1x protease inhibitor cocktail, 250 nM ADP-HPD, and 10 µM PJ34) and incubated on ice for 30 min. The nuclear fraction was collected by centrifugation (12,000 x g, 30 min, 4°C). The nuclear and cytoplasmic extracts were collected, aliquoted, flash frozen in liquid nitrogen, and stored at −80°C until used.

## Western blotting

The protein concentrations of the whole cell lysates were determined using Bio-Rad Protein Assay Dye Reagent (Bio-Rad, 5000006). Volumes of lysates containing equal total amounts of protein were boiled at 100°C for 5 min after addition of 1/4 vol of 4x SDS-PAGE Loading Solution (250 mM Tris pH 6.8, 40% glycerol, 0.04% Bromophenol blue, 4% SDS), run on 10% polyacrylamide-SDS gels, and transferred to nitrocellulose membranes. After blocking with 5% nonfat milk in TBST, the membranes were incubated with the primary antibodies described above in TBST with 0.02% sodium azide, followed by anti-rabbit HRP-conjugated IgG (1:5000) or anti-mouse HRP-conjugated IgG (1:5000). Immunoblot signals were detected using an ECL detection reagent (Thermo Fisher Scientific, 34577, 34095).

## Cell growth assays

Cells were transfected with control or *PARP7*-specific siRNAs at a final concentration of 30 nM in 10 cm diameter culture dishes. Twenty-four hours later, the cells were collected and plated at a density of 20,000 cells per well in a 24-well plate. The cells were fixed with 4% paraformaldehyde at each of the indicated time points and washed with water. The plates were stored at 4°C until the end of the time course when all samples could be analyzed simultaneously. After all samples were collected, the fixed cells were stained with crystal violet (0.5% crystal violet in 20% methanol) for 30 min with gentle shaking at room temperature. The stained cells were washed several times with water and air dried. The crystal violet in each well was re-dissolved in 10% acetic acid and the absorbance at 570 nm was read using a spectrophotometer. The absorbance of a blank well was subtracted from the samples and the values were normalized to the values at day 1. Three independent experiments were performed with independent biological replicates to ensure reproducibility. Statistical differences were determined using Student's t-tests at each time point.

## Cell migration and invasion assays

Boyden chamber assays were used to determine the migration and invasive capacity of cells as described below. The cells cultured in six-well plates were transfected with 30 nM control siRNA or two different siRNAs targeting *PARP7* mRNA. Twenty-four hours later, the cells were trypsinized and were seeded at a density of 100,000 cells into migration chambers (Corning, 353097) or invasion chambers (Corning, 354480) following the manufacturer's protocols. Briefly, the trypsinized cells were resuspended in serum-free RPMI media and collected by centrifugation at 300 x g for 3 min at room temperature. The cell pellets were resuspended in serum-free RPMI media and 500 µL of the media containing 100,000 cells were plated into the top chamber. The chambers were then incubated in 750 µL of RPMI media with 10% FBS. For rescue experiments in OVCAR4 cells expressing wild-type or catalytically dead mutant PARP-7, the cells were plated as above and then treated with 1 µg/mL doxycycline (Dox). After 24 hr of incubation at 37°C, the cells in the top chamber were scraped and removed. The chambers were stained with crystal violet (0.5% crystal violet in 20% methanol) for 30 min with gentle shaking. The chambers were washed with water, air-dried and the images of cells at the bottom of the membrane were collected using an upright microscope. Three independent biological replicates were performed for each condition. Statistical differences were determined using Student's t-test.

## RNA isolation and reverse transcription-quantitative real-time PCR (RT-qPCR)

OVCAR4 cells were plated in 6-well plates and transfected with 30 nM of *PARP7* or control siRNAs as described above. Total RNA was isolated using the Qiagen RNAeasy Plus Mini kit (Qiagen, 74136) according to the manufacturer's protocol. Total RNA was reverse transcribed using oligo(dT) primers and MMLV reverse transcriptase (Promega, PR-M1705) to generate cDNA. The cDNA

samples were subjected to RT-qPCR using gene-specific primers, as described below. Target gene expression was normalized to the expression of *RPL19* mRNA. All experiments were repeated a minimum of three times with independent biological samples to ensure reproducibility and a statistical significance of at least p<0.05. Statistical differences between control and experimental samples were determined using Student's t-test. The primers used for RT-qPCR were as follows:

- *RPL19* forward: 5'- ACATCCACAAGCTGAAGGCA-3'
- *RPL19* reverse: 5'- TGCGTGCTTCCTTGGTCTTA −3'
- *PARP7* forward: 5'- CCAAAACCAGTTTCTTTGGGAG −3'
- *PARP7* reverse: 5'- CAGATTCCATCTACCACATCC −3'

## RNA-sequencing and data analyses

### Generation of RNA-seq libraries

Two biological replicates of total RNA were isolated from each siRNA-mediated knockdown. Total RNA was isolated using the RNeasy kit (Qiagen, 74106) according to the manufacturer's instructions. The total RNA was then enriched for polyA+ RNA using Dynabeads Oligo(dT)25 (Invitrogen). The polyA+ RNA was then used to generate strand-specific RNA-seq libraries as described previously (*Zhong et al., 2011*). The RNA-seq libraries were subjected to QC analyses (final library yield, and the size distribution of the final library DNA fragments) and sequenced using an Illumina HiSeq 2000.

### Analysis of RNA-seq data

The raw data were subjected to QC analyses using the FastQC tool (*Andrews, 2010*). The reads were then mapped to the human genome (hg38) using the spliced read aligner TopHat, version.2.0.13 (*Kim et al., 2013*). Transcriptome assembly was performed using cufflinks v.2.2.1 (*Trapnell et al., 2010*) with default parameters. The transcripts were merged into two distinct, non-overlapping sets using cuffmerge (*Trapnell et al., 2010*), followed by cuffdiff (*Trapnell et al., 2010*) to call the differentially regulated transcripts. The significantly regulated genes (p<0.05) in the *PARP7* knockdown samples compared to the control samples were integrated to find the commonly regulated gene set.

### Transcriptome data analyses

We obtained transcript abundance in terms of FPKM for all transcripts using cuffdiff (*Trapnell et al., 2010*). A fold change cutoff of 1.5-fold for upregulated genes and –0.5-fold for downregulated genes was applied to all samples to limit our focus on those transcripts whose expression was clearly affected by *PARP7* knockdown. The genes passing the filters were used to make heat maps using Java TreeView (*Saldanha, 2004*).

### Gene ontology analyses

We performed gene ontology analyses using the Database for Annotation, Visualization, and Integrated Discovery (DAVID) Bioinformatics Resources tool for gene ontology analysis (*Huang et al., 2009a*; *Huang et al., 2009b*) for genes specifically enriched in the *PARP7* knockdown samples compared to the control samples based on expression levels. The inputs used were genes either upregulated or downregulated in *PARP7* knockdown samples, generating two sperate gene ontology plots.

## Structure-based alignment of NAD⁺ analog-sensitive 'gatekeeper' residues in PARP-7

Multiple sequence alignment of PARP-7 (Q7Z3E1), PARP-1 (P09874), and PARP-12 (Q9H0J9) was performed using Clustal Omega (*Madeira et al., 2019*). The catalytic domains of all three PARPs, especially the two examined homologous gatekeeper residues, are highly conserved. Available structures of PARP-1 (PDBID: 3PAX) and PARP-12 (PDBID: 2PQF) were downloaded from the RCSB Protein Data Bank for analysis. The molecular graphic was generated using PyMOL software (PyMOL Molecular Graphics System, Version 2.3.2) (*Schrodinger LLC, 2010*). A close-up view of residues F547 and S563 in PARP-7 were illustrated by comparison of the available structures of PARP-1 and PARP-12.

## Expression and purification of wild-type and mutant PARP-7 proteins

### Cloning of a PARP7 cDNA

cDNA pools from OVCAR4 cells were prepared by extraction of total RNA using Trizol reagent (ThermoFisher Scientific, 15596018), followed by reverse transcription using superscript III reverse transcriptase (ThermoFisher Scientific, 18080–093) and an oligo(dT) primer according to manufacturer's instructions. The human *PARP-7* cDNA was then amplified by PCR from the cDNA pools described above, adding sequences encoding an N-terminal Flag epitope to the cDNAs using the primers below.

### Site-directed mutagenesis of the PARP7 cDNA

Base pair alterations changing the codon for phenylalanine 547 (F547) to glycine (G), alanine (A), or valine (V), or the codon for serine 563 (S563) to G, A, V, threonine (T), or isoleucine (I) were introduced into the *PARP-7* cDNA by PCR-based site-directed mutagenesis. The F547A mutant *PARP-7* cDNA was subjected to additional site-directed mutagenesis to generate double mutants by altering the codon for S563 to V, T or I individually. The primers for site-mutagenesis are listed below.

### Generation of insect cell expression vectors and recombinant baculoviruses

The PCR products were digested using restriction enzymes *BamHI* and *EcoRI*, and ligated into *BamHI-* and *EcoRI*-digested pFastBac1 plasmid. All constructs were sequenced to ensure the fidelity of the sequences. The recombinant pFastBac1 bacmids were then prepared for transfection into Sf9 cells by transformation into the DH10BAC *E. coli* strain with subsequent blue/white colony screening using the Bac-to-Bac system (Invitrogen) according to the manufacturer's instructions.

Primers used for PARP-7 cDNA cloning:

- *PARP-7* forward: 5'-TTAACCGGATCCATGGACTACAAAGACGATGACGACAAGAGCGAAATGGAAACCACCGAACCTGAGCCAG-3'
- *PARP-7* reverse: 5'-CCTTCCGAATTCTTATCAAATGGAAACAGTGTTACTGACTTCTTCATATTGGATAAC-3'

Primers used for site-mutagenesis:

- *PARP-7 F547G* forward: 5'-CAGGATGTGGTAGATGGAATCTGCAAACACAACGGTGACCCTCGAGTCTGTGGAAAGCATGC-3'
- *PARP-7 F547G* reverse: 5'-GCATGCTTTCCACAGACTCGAGGGTCACCGTTGTGTTTGCAGATTCCATCTACCACATCCTG-3'
- *PARP-7 F547A* forward: 5'-CAGGATGTGGTAGATGGAATCTGCAAACACAACGCTGACCCTCGAGTCTGTGGAAAGCATGC-3'
- *PARP-7 F547A* reverse: 5'-GCATGCTTTCCACAGACTCGAGGGTCAGCGTTGTGTTTGCAGATTCCATCTACCACATCCTG-3'
- *PARP-7 F547V* forward: 5'-CAGGATGTGGTAGATGGAATCTGCAAACACAACGTTGACCCTCGAGTCTGTGGAAAGCATGC-3'
- *PARP-7 F547V* reverse: 5'-GCATGCTTTCCACAGACTCGAGGGTCAACGTTGTGTTTGCAGATTCCATCTACCACATCCTG-3'
- *PARP-7 S563G* forward: 5'-GCTACAATGTTTGGACAAGGCGGTTATTTTGCAAAGAAGGC-3'
- *PARP-7 S563G* reverse: 5'-GCCTTCTTTGCAAAATAACCGCCTTGTCCAAACATTGTAGC-3'
- *PARP-7 S563A* forward: 5'-GCTACAATGTTTGGACAAGGCGCTTATTTTGCAAAGAAGGC-3'
- *PARP-7 S563A* reverse: 5'-GCCTTCTTTGCAAAATAAGCGCCTTGTCCAAACATTGTAGC-3'
- *PARP-7 S563V* forward: 5'-GCTACAATGTTTGGACAAGGCGTTTATTTTGCAAAGAAGGCAAGC-3'
- *PARP-7 S563V* reverse: 5'-GCTTGCCTTCTTTGCAAAATAAACGCCTTGTCCAAACATTGTAGC-3'
- *PARP-7 S563T* forward: 5'-GCTACAATGTTTGGACAAGGCACTTATTTTGCAAAGAAGGCAAGE-3'
- *PARP-7 S563T* reverse: 5'-GCTTGCCTTCTTTGCAAAATAAGTGCCTTGTCCAAACATTGTAGC-3'
- *PARP-7 S563I* forward: 5'-GCATGCTACAATGTTTGGACAAGGCATTTATTTTGCAAAGAAGGCAAGC-3'
- *PARP-7 S563I* reverse: 5'-GCTTGCCTTCTTTGCAAAATAAATGCCTTGTCCAAACATTGTAGCATGC-3'

## Purification of PARP-7 expressed in Sf9 insect cells

Sf9 insect cells were cultured in SF-II 900 medium (Invitrogen) and transfected with 1 μg of recombinant bacmids for expressing Flag-tagged wild-type (WT) or analog-sensitive mutant human PARP 7 (asPARP-7) using Cellfectin transfection reagent (Invitrogen) according to the manufacturer's instructions. After 3 days, the medium was supplemented with 10% FBS, penicillin, and streptomycin, and collected as a baculovirus stock. After multiple rounds of amplification of the stock, the resulting high titer baculovirus was used to infect fresh Sf9 cells to induce PARP-7 protein expression for 2 days. The PARP-7-expressing Sf9 cells were collected by centrifugation, flash frozen in liquid $N_2$, and stored at −80°C until used.

For purification of PARP-7 proteins, the Sf9 cell pellets were thawed on ice, resuspended in Flag Lysis Buffer [50 mM HEPES pH 7.9, 0.5 M NaCl, 4 mM $MgCl_2$, 0.4 mM EDTA, 20% glycerol, 5 mM β-mercaptoethanol, 2x protease inhibitor cocktail (Roche)], and lysed by Dounce homogenization (Wheaton). The lysate was clarified by centrifugation, mixed with an equal volume of Flag Dilution Buffer (50 mM HEPES pH 7.9, 10% glycerol, 0.02% NP-40), sonicated, and then clarified by centrifugation again. The clarified lysate was mixed with anti-Flag M2 agarose resin (Sigma), washed twice with Flag Wash Buffer #1 (50 mM HEPES pH 7.9, 150 mM NaCl, 2 mM $MgCl_2$, 0.2 mM EDTA, 15% glycerol, 0.01% NP-40, 5 mM β-mercaptoethanol, 1 mM PMSF, 1 μM aprotinin, 100 μM leupeptin), twice with Flag PARP Wash Buffer #2 (50 mM HEPES pH 7.9, 1 M NaCl, 2 mM $MgCl_2$, 0.2 mM EDTA, 15% glycerol, 0.01% NP-40, 5 mM β-mercaptoethanol, 1 mM PMSF, 1 μM aprotinin, 100 μM leupeptin), and twice with Flag PARP Wash Buffer #3 (50 mM HEPES pH 7.9, 150 mM NaCl, 2 mM $MgCl_2$, 0.2 mM EDTA, 15% glycerol, 0.01% NP-40, 5 mM β-mercaptoethanol, 1 mM PMSF). The Flag-tagged PARP-7 proteins were then eluted from the anti-Flag M2 agarose resin with Flag Wash Buffer #3 containing 0.2 mg/mL 3x Flag peptide (Sigma). The eluted proteins (~0.5 mg/mL) were distributed into single use aliquots, flash frozen in liquid $N_2$, and stored at −80°C until use.

## NAD⁺ analogs

The $NAD^+$ analogs used in this study were either purchased from, or synthesized in collaboration with, the BIOLOG Life Science Institute, Bremen, Germany (*Gibson et al., 2016*). All the $NAD^+$ analogs are listed in *Supplementary file 2*.

## In vitro PARP-7 autoMARylation reactions

### PARP-7 autoMARylation reactions

One microgram of purified recombinant PARP-7 proteins (wild-type or mutant) were incubated in ADP-ribosylation Reaction Buffer [30 mM HEPES pH 8.0, 5 mM $MgCl_2$, 5 mM $CaCl_2$, 0.01% NP-40, 1 mM DTT, 100 ng/μL sonicated salmon sperm DNA (Stratagene), and 100 ng/μL BSA (Sigma)] with 250 μM $NAD^+$ or $NAD^+$ analogs at room temperature for 30 min.

### Detection of PARP-7 autoMARylation by western blotting

The autoADPRylation reaction described above was stopped by the addition of one third of a reaction volume of 4x SDS-PAGE Loading Buffer (200 mM Tris-HCl pH 6.8, 8% SDS, 40% glycerol, 4% β-mercaptoethanol, 50 mM EDTA, 0.08% bromophenol blue) followed by incubation at 100°C for 10 min. The reactions were then loaded onto a 10% PAGE-SDS gel and transferred to a nitrocellulose membrane. After blocking with 5% nonfat milk in TBST, the membrane was then blotted with antibodies against PARP-7 (Invitrogen) or Flag (Invitrogen), or an ADP-ribose detection reagent (MABE1016, EMD Millipore), followed by blotting with anti-rabbit HRP-conjugated IgG (1:5000) or anti-mouse HRP-conjugated IgG (1:5000) secondary antibodies. Immunoblot signals were detected using an ECL detection reagent (Thermo Fisher Scientific, 34577, 34095), and quantified using Image Lab 6.0 (Bio-Rad).

### Detection of PARP-7 autoMARylation by in-gel fluorescence

The autoMARylation reactions were stopped by methanol:chloroform precipitation with subsequent collection of the precipitates by centrifugation. The protein pellets were dissolved and 'clicked' to azido-rhodamine (Click Chemistry Tools) in Denaturing CC Buffer (100 mM HEPES pH 8.0, 4 M urea,

0.5 M NaCl, 2% CHAPS, 100 µM azido-rhodamine, 5 mM THPTA [Click Chemistry Tools], 1 mM CuSO$_4$, 5 mM sodium ascorbate) in the dark at room temperature for 2 hr. The clicked proteins were collected by methanol:chloroform precipitation with centrifugation, dissolved in 1x SDS Loading Buffer (50 mM Tris-HCl, pH 6.8, 2% SDS, 10% glycerol, 1% β-mercaptoethanol, 12.5 mM EDTA, 0.02% bromophenol blue), and incubated at 100°C for 10 min in the dark. The clicked proteins were run on a 10% PAGE-SDS gel in the dark. After washing twice with 10% methanol and twice with MilliQ H$_2$O, the gel was then visualized on a Bio-Rad Pharos FX Plus Molecular Imager (excitation: 532 nm, emission: 605 nm) (*Gibson and Kraus, 2017*; *Gibson et al., 2016*).

## Determination of relative PARP-7 activity with NAD$^+$ and NAD$^+$ analogs

PARP-7 automodification in the in vitro assays was determined by western blotting as described above. Immunoblot signals were detected by ECL and quantified using Image Lab 6.0 (Bio-Rad). For determination of relative PARP-7 activity with NAD$^+$ or each NAD$^+$ analog, the signal from a 'water only control' run simultaneously with the experimental conditions was subtracted from itself and each PARP-7 automodification signal (wild-type or mutant, NAD$^+$ or NAD$^+$ analog). The signal from wild-type PARP-7 with NAD$^+$ was then set to 100, The water control had a value of 0 because of the subtraction of the background signal from itself. Any negative values after the water background subtraction (i.e. less than background) were set to zero. The signals from the wild-type or mutant PARP-7 proteins with NAD$^+$ or the NAD$^+$ analogs were then expressed relative to the background-corrected signal from wild-type PARP-7 and NAD$^+$. This yielded values that ranged from 0 to more than 100, with the highest values coming from pairings between PARP-7 mutants and NAD$^+$ analogs that yielded more PARP-7 automodification than wild-type PARP-7 with NAD$^+$.

## Identification of PARP-7 substrates in OVCAR4 cells using an analog-sensitive approach

### In-gel fluorescence of small-scale PARP-7-specific modification of OVCAR4 whole cell extract proteins

OVCAR4 cytosolic S100 extract was prepared as previously described (*Dignam et al., 1983*; *Mayeda and Krainer, 1999*). One microgram of PARP-7 protein (wild-type or analog-sensitive mutant) was incubated with 50 µg of OVCAR4 whole cell extract and 250 µM NAD$^+$ or 8-Bu(3-yne)T-NAD$^+$ in ADP-ribosylation Reaction Buffer (see above) at room temperature for 30 min. The reactions were stopped by methanol:chloroform precipitation with subsequent collection of the precipitates by centrifugation. The protein pellets were 'clicked' to azido-rhodamine (Click Chemistry Tools) in Denaturing CC Buffer, run on an SDS-PAGE gel, and visualized using a Bio-Rad Pharos FX Plus Molecular Imager, as described above.

### Large-scale PARP-7-specific modification of OVCAR4 and HeLa cell cytosolic extract proteins

Twenty micrograms of analog-sensitive PARP-7 protein (wild-type or asPARP-7 mutant) were incubated sequentially with 1 mg of OVCAR4 cell or HeLa cell cytosolic extract in ADP-ribosylation Reaction Buffer at room temperature for 5 min and then after the addition of 250 µM 8-Bu(3-yne)T-NAD$^+$ for 30 min. The reactions were stopped by methanol:chloroform precipitation. The protein pellets were resuspended in 1 mL of Urea Solubilization Buffer (200 mM HEPES pH 8.0, 8 M urea, 1 M NaCl, 4% CHAPS) and the insoluble proteins were pelleted by centrifugation for 1 min at maximum speed in a microcentrifuge.

Proteins soluble in the Urea Solubilization Buffer were combined sequentially in a 2 mL tube with the following: 100 µL azido-agarose Beads (Click Chemistry Tools), 820 µL water, 40 µL of a 50:250 mM CuSO4:THPTA pre-formed catalytic complex, 20 µL of 500 mM aminoguanidine hydrochloride, and 20 µL of 500 mM sodium ascorbate. The reaction was incubated in the dark on a rotating mixer for 18 hr at room temperature. The beads were then collected by centrifugation at room temperature for 1 min at 1,000 RCF in a microcentrifuge and the reaction supernatant was aspirated. The beads were resuspended in 1.8 mL MilliQ H$_2$O and were collected by centrifugation at room temperature for 1 min at 1000 RCF in a microcentrifuge.

The beads were then suspended in 1 mL of SDS Wash Buffer (100 mM Tris-HCl pH 8.0, 1% SDS, 250 mM NaCl, 5 mM EDTA) with the addition of freshly made 1 mM DTT. The reaction was then heated to 70°C for 15 min and allowed to cool to room temperature. The reaction was then centrifuged at room temperature for 5 min at 1000 RCF in a microcentrifuge and the supernatant was aspirated. The resin was then resuspended in 1 mL of SDS Wash Buffer containing 40 mM iodoacetamide and incubated at room temperature for 30 min in the dark to alkylate the cysteine residues. The resin was then transferred to a 2 mL single use column (Bio-Rad) and washed as follows in sequential order: 10 washes with 2 mL of SDS Wash Buffer (see above), 10 washes with 2 mL of Urea Wash Buffer (100 mM Tris•HCl, pH 8.0, 8M urea), and 10 washes with 2 mL of 20% acetonitrile.

### On-bead trypsin digestion

The resin was resuspended in 500 µL of Trypsin Digestion Buffer (100 mM Tris·HCl pH 8.0, 2 mM CaCl$_2$, 10% acetonitrile). Trypsin digestion of bead-bound 8-Bu(3-yne)T-ADP-ribosylated proteins was performed by adding of 1 µg of trypsin (Promega) to the Trypsin Digestion Buffer, with incubation at room temperature overnight on a rotating mixer. The peptides from the tryptic digest were collected and lyophilized for storage at −80°C prior to LC-MS/MS analysis.

### On-bead hydroxylamine treatment

After the trypsin digestion of the proteins on the beads, which left the peptides covalently linked to the beads through the azide-clicked 8-Bu(3-yne)T-ADP-ribosylation sites, the beads were transferred to a fresh 2 mL single use column (Bio-Rad) and washed as follows: 10 washes of 2 mL each with SDS Wash Buffer, 10 washes of 2 mL each with Urea Wash Buffer, 10 washes of 2 mL each with 20% acetonitrile, and 5 washes of 2 mL each with Peptide Elution Buffer (100 mM HEPES, pH 8.5). The beads were transferred to a microcentrifuge tube and hydroxylamine (Sigma) was added to 0.5 M to elute the glutamate- and aspartate-modified 8-Bu(3-yne)T-ADP-ribosylated peptides from the beads. The eluted peptides were prepared for LC-MS/MS by desalting on a C18 stage tip (Thermo) according to the manufacturer's protocol and then lyophilized for storage at −80°C prior to LC-MS/MS analysis.

### LC-MS/MS analysis of trypsin- and hydroxylamine-eluted samples

Following solid-phase extraction cleanup with an Oasis HLB microelution plate (Waters), the trypsin samples were reconstituted in 2% (v/v) acetonitrile (ACN) and 0.1% trifluoroacetic acid in water such that the resulting concentration was 1 µg/µL. Hydroxylamine-eluted samples were reconstituted in 6 µL of the same buffer. One microliter of the trypsin digest samples were injected onto an Orbitrap Fusion Lumos mass spectrometer (Thermo Electron) coupled to an Ultimate 3000 RSLC-Nano liquid chromatography system (Dionex). Samples were injected onto a 75 µm i.d., 50 cm long EasySpray column (Thermo) and eluted with a gradient from 1–28% buffer B over 60 min. Buffer A contained 2% (v/v) ACN and 0.1% formic acid in water, and Buffer B contained 80% (v/v) ACN, 10% (v/v) trifluoroethanol, and 0.1% formic acid in water.

The mass spectrometer was operated in positive ion mode with a source voltage of 1.5–2.4 kV and an ion transfer tube temperature of 275°C. MS scans were acquired at a resolution of 120,000 in the Orbitrap and up to 10 MS/MS spectra were obtained in the ion trap for each full spectrum acquired using higher-energy collisional dissociation (HCD) for ions with charges 2–7. Dynamic exclusion was set for 25 s after an ion was selected for fragmentation. Raw MS data files were analyzed using Proteome Discoverer v2.2 (Thermo), with peptide identification performed using Sequest HT searching against the Human database from UniProt. Fragment and precursor tolerances of 10 ppm and 0.6 Da were specified, and three missed cleavages were allowed. Carbamidomethylation of cysteine was set as a fixed modification, with oxidation of Met and hydroxamic acid addition (+15.0109 Da) to Asp and Glu set as variable modifications. The false-discovery rate (FDR) cutoff was 1% for all proteins and peptides.

### Assignment of ADPRylation sites in α-tubulin

The ADPRylation site calls from the mass spectra for α-tubulin (see *Supplementary file 5*) were manually mapped to the UniProt Q1ZYQ1 (human tubulin alpha chain) protein.

### Analysis and representation of the mass spectrometry data

The peptides identified from the mass spectrometry experiments were used for further analyses and the data were expressed in various formats, as described below.

#### Venn diagrams

The peptide identifications (IDs) in each of the replicates for the two cell lines (OVCAR4 and HeLa) were used to generate a Venn diagram of overlapping peptide IDs using jvenn software (*Bardou et al., 2014*). The universe of peptide IDs from replicates of each cell line were then used to make a final Venn diagram of the combined peptides from the two cell lines, collapsed into individual proteins. The common protein IDs from the two cell lines was used for gene ontology.

#### Gene ontology analysis

Gene ontology analyses were performed using the DAVID online bioinformatics resource (*Dennis et al., 2003*) for the common protein IDs from the two cell lines as described above.

### Immunoprecipitation of PARP-7 substrate proteins

Immunoprecipitation (IP) of MARylated proteins from OVCAR4 cells was performed as follows. OVCAR4 cells plated on 15 cm diameter cell culture dishes were transfected with 30 nM control siRNA or two different siRNAs targeting PARP-7. Two 15 cm dishes of cells were used for each siRNA transfection. Forty-eight hours after transfection, the cells were collected, washed twice with ice-cold PBS, and resuspended in IP Lysis Buffer (50 mM Tris-HCl pH7.5, 0.5 M NaCl, 1.0 mM EDTA, 1% NP-40 and 10% glycerol, freshly supplemented with 1 mM DTT, 250 nM ADP-HPD, 10 μM PJ34, 1x complete protease inhibitor cocktail), and incubated at 4°C for 30 min with gentle shaking. The cell debris was cleared by centrifugation for 15 min at 4°C at 15,000 g. The supernatants were collected and the protein concentrations were measured using a Bradford assays.

Aliquots of cell lysates containing equal amount of protein were used for each IP condition. Five percent of each cell lysate was saved for input. The cell lysates for IP were incubated with 3 μg of MAR detection reagent or rabbit IgG (Thermo Fisher Scientific, 10500C), and protein A agarose beads overnight at 4°C with gentle rotation. The beads were then washed five times with Lysis Buffer for 5 min each at 4°C with gentle mixing. The beads were then heated to 100°C for 10 min in 1x SDS-PAGE loading buffer to release the bound proteins. The immunoprecipitated material was subjected to western blotting as described above.

### Generation of vectors for the expression of wild-type and MARylation site mutant α-tubulin

The plasmid for expression of C-terminal HA-tagged α-tubulin was obtained from Sino Biologicals (HG14201-CY). Mutations of the sites of MARylation (i.e. D69, E71 and E77) were introduced into the pCMV3-C-HA-TUBA3C plasmid using a protocol adapted from the Quickchange site-directed mutagenesis kit (Agilent). D69N and E71Q were added first, then E77Q to the double mutant to generate the triple mutant using the primers listed below.

Primers used for generatingα-tubulin MARylation site mutants:

- *α-tubulin D69N/E71Q forward:* 5'- GAGCAGTGTTTGTGAACCTGCAGCCCACTG −3'
- *α-tubulin D69N/E71Q reverse:* 5'- CAGTGGGCTGCAGGTTCACAAACACTGCTC −3'
- *α-tubulin D69N/E71Q/E77Q forward:* 5'- GTGAACCTGCAGCCCACTGTGGTCGATCAAGTG −3'
- *α-tubulin D69N/E71Q/E77Q reverse:* 5'- CACTTGATCGACCACAGTGGGCTGCAGGTTCAC −3'

### Immunoprecipitation of HA-tagged α-tubulin

Immunoprecipitation (IP) of HA-tagged α-tubulin from OVCAR4 or 293 T cells was performed as follows:

For Nocodazole or cold treatments, OVCAR4 cells were transiently transfected with an expression vector for HA-tagged α-tubulin. Forty-eight hours after transfection, the cells were either placed on ice for 45 min or treated with 6 μM nocodazole for 1 hr and then washed with warm medium three

times. The cells were allowed to recover by incubation in normal growth medium at 37°C for 15 min, and then collected and extracted as described above.

For the induction of PARP-7 expression in OVCAR4 cells, cells expressing wild-type or catalytically dead mutant PARP-7 were transiently transfected with HA-tagged α-tubulin and simultaneously treated with 1 μg/mL Dox. For expression of PARP-7 in 293 T cells, the cells were transiently transfected with Flag-tagged PARP-7 and wild-type or mutant α-tubulin constructs. Forty-eight hours after transfection, the cells were collected as described above.

Cell lysates for IP were incubated with 1.5 μg of mouse monoclonal antibody against HA-tag and protein G agarose beads overnight at 4°C with gentle rotation. The beads were then washed three times with Lysis Buffer for 5 min each at 4°C with gentle mixing. The beads were then heated to 100°C for 5 min in 1x SDS-PAGE loading buffer to release the bound proteins. The immunoprecipitated material was subjected to western blotting as described above.

## Generation of lentiviral expression vectors for wild-type and catalytically dead mutant (Y564A) mouse PARP-7

The *Parp7* cDNA was prepared by extracting total RNA from 3T3-L1 cells using Trizol (Invitrogen, 15596026), followed by reverse transcription using Superscript III reverse transcriptase (Invitrogen, 18080051) and an oligo(dT) primer, according to the manufacturer's instructions. The *Parp7* cDNA was then amplified from the cDNA library with primers listed below and cloned into pCDNA3 using the primers listed below. A cDNA for the catalytically dead PARP-7 mutant (Y564A) was generated by site-directed mutagenesis using Pfu Turbo DNA polymerase (Agilent, 600250) using the primers listed below. The PCR products were then amplified with primers encoding an N-terminal FLAG epitope tag and cloned into the pINDUCER20 lentiviral Dox-inducible expression vector (Addgene, 44012).

Primers used for amplifying Parp7 cDNA:

- *PARP-7* forward: 5'- ATGGATTATAAGGATGACGATGACAAGATGGAAGTGGAAACCACTG −3'
- *PARP-7* reverse: 5'- GTTTAATTAATCATTACTACTTAAATGGAAACAGTGTTACTG −3'

Primers used for cloning Parp7 cDNA into pCDNA3 vector:

- *PARP-7* forward: 5'- CTGGCTAGCGTTTAAACTTAATGGATTATAAGGATGACG −3'
- *PARP-7* reverse: 5'- AGCGGGTTTAAACGGGCCCTTTAAATGGAAACAGTGTTACTG −3'

Primers used for cloning Parp7 cDNA into pINDUCER20 vector:

- *PARP-7* forward: 5'- TCCGCGGCCCCGAACTAGTGATGGATTATAAGGATGACG −3'
- *PARP-7* reverse: 5'- GTTTAATTAATCATTACTACTTAAATGGAAACAGTGTTACTG −3'

Primers used for generating Parp7 catalytic dead mutant, Y564A:

- *Parp7 Y564A* forward: 5'- AGCTTGCCTTCTTTGCGAAAGCACTGCCTTGTCCAAACATTG −3'
- *Parp7 Y564A* reverse: 5'- CAATGTTTGGACAAGGCAGTGCTTTCGCAAAGAAGGCAAGCT −3'

## Generation of cell lines with inducible ectopic expression of PARP-7

Cells were transduced with lentiviruses for Dox-inducible ectopic expression of PARP-7 (wild-type and mutant). We generated lentiviruses by transfection of the pINDUCER20 constructs described above, together with: (i) an expression vector for the VSV-G envelope protein (pCMV-VSV-G, Addgene 8454), (ii) an expression vector for GAG-Pol-Rev (psPAX2, 12260), and (iii) a vector to aid with translation initiation (pAdVAntage, Promega) into 293 T cells using Lipofectamine 3000 reagent (Invitrogen, L3000015) according to the manufacturer's instructions. The resulting viruses were collected in the culture medium, concentrated by using a Lenti-X concentrator (Clontech, 631231), and used to infect OVCAR4 cells. Stably transduced cells were selected with G418 sulfate (Sigma, A1720; 1 mg/ml). The cells were treated with 1 μg/mL Dox for 24 hr to induce protein expression. Inducible ectopic expression of the cognate proteins was confirmed by western blotting.

## Immunofluorescent staining of cultured cells

OVCAR4 cells grown in six-well plates were transfected with 30 nM control siRNA or two different siRNAs targeting PARP-7. Twenty-four hours after transfection, the cells were trypsinized and seeded into eight-well chambered slides (Thermo Fisher, 154534). For rescue experiments in OVCAR4 cells expressing wild-type or catalytically dead mutant PARP-7, the cells were plated as above and then treated with 1 µg/mL Dox. Twenty-four hours later, the cells were either placed on ice for 45 min or treated with 6 µM nocodazole for 1 hr and then washed with warm medium. The cells were allowed to recover by incubation in normal growth medium at 37°C for 15 min and were then washed twice with PBS, fixed with 4% paraformaldehyde for 15 min at room temperature, and washed three times with PBS. The cells were permeabilized for 5 min using Permeabilization Buffer (PBS containing 0.01% Triton X-100), washed three times with PBS, and incubated for 1 hr at room temperature in Blocking Solution (PBS containing 1% BSA, 10% FBS, 0.3 M glycine and 0.1% Tween-20). The fixed cells were incubated with α-tubulin antibody (1:1000 Santa Cruz, sc-8035) or HA antibody (1:200 Abcam, ab9110) in PBS overnight at 4°C. After washing three times with PBS, the cells were incubated with Alexa Fluor 488 goat anti-mouse IgG (1:500, ThermoFisher, A-11001) in PBS for 1 hr at room temperature. After washing three times with PBS, coverslips were mounted with the VectaShield Antifade Mounting Medium with DAPI (Vector Laboratories, H-1200). All images were acquired using an inverted Zeiss LSM 780 confocal microscope and the fluorescence intensities were measured using Fiji ImageJ software.

## Acknowledgements

We thank members of the Kraus lab for critical comments on this manuscript. We also thank Lianying Jiao for the structure-based alignment of the gatekeeper residues in PARP-7 and Michael Cohen for sharing his lab's results on PARP-7 prior to publication and providing helpful suggestion on our work. We acknowledge and thank the following UT Southwestern core facilities: the Next Generation Sequencing Core for deep sequencing services (Dr. Ralf Kittler and Vanessa Schmid), the Proteomics Core Facility for mass spectrometry services (Dr. Hamid Mirzaei), and the Live Cell Imaging Core for microscopy support (Dr. Katherine Luby-Phelps).

## Additional information

### Competing interests

Bryan A Gibson: holds the patents on the anti-MAR binding reagent (United States Patent No. 9,599,606) and the asPARP technology (United States Patent No. 9,926,340) described herein. UT Southwestern Medical Center has licensed the anti-MAR binding reagent to EMD Millipore, which markets it for research purposes. BIOLOG Life Science Institute, a coholder of United States Patent No. 9,926,340, sells the NAD+ analog 8-Bu(3-yne)T-NAD+. W Lee Kraus: is a founder consultant for Ribon (Therapeutics, Inc). Holds the patents on the anti-MAR binding reagent (United States Patent No. 9,599,606) and the asPARP technology (United States Patent No. 9,926,340) described herein. UT Southwestern Medical Center has licensed the anti-MAR binding reagent to EMD Millipore, which markets it for research purposes. BIOLOG Life Science Institute, a coholder of United States Patent No. 9,926,340, sells the NAD+ analog 8-Bu(3-yne)T-NAD+. The other authors declare that no competing interests exist.

### Funding

| Funder | Grant reference number | Author |
| --- | --- | --- |
| National Institute of Diabetes and Digestive and Kidney Diseases | R01 DK069710 | W Lee Kraus |

The funders had no role in study design, data collection and interpretation, or the decision to submit the work for publication.

## Author contributions

Lavanya H Palavalli Parsons, Conceptualization, Data curation, Formal analysis, Validation, Investigation, Methodology, Writing - original draft; Sridevi Challa, Data curation, Formal analysis, Investigation, Writing - original draft; Bryan A Gibson, Conceptualization, Data curation, Formal analysis, Validation, Investigation, Methodology; Tulip Nandu, Data curation, Formal analysis, Investigation, Visualization, Writing - original draft; MiKayla S Stokes, Investigation, Writing - original draft; Dan Huang, Formal analysis, Investigation, Visualization, Writing - original draft, Writing - review and editing; Jayanthi S Lea, Resources, Data curation, Investigation, Project administration; W Lee Kraus, Conceptualization, Formal analysis, Supervision, Funding acquisition, Investigation, Visualization, Writing - original draft, Project administration, Writing - review and editing

## Author ORCIDs

MiKayla S Stokes http://orcid.org/0000-0002-2144-4343
W Lee Kraus https://orcid.org/0000-0002-8786-2986

## Decision letter and Author response

Decision letter https://doi.org/10.7554/eLife.60481.sa1
Author response https://doi.org/10.7554/eLife.60481.sa2

# Additional files

**Supplementary files**

• Supplementary file 1. RNA-seq analysis of gene expression in ovarian cancer cells following PARP-7 depletion. RNA-seq was performed after siRNA-mediated knockdown (KD) of PARP7 in OVCAR4 cells. Three different siRNAs were used versus a control siRNA. The table lists the expression values in FPKM obtained using the Cufflinks pipeline described in the Methods for all the genes across all the samples for the RNA-seq data.

• Supplementary file 2. $NAD^+$ analogs used in this study. The synthesis of many of these $NAD^+$ analogs was reported in *Gibson et al., 2016*. Most of the compounds can be purchased from the BIOLOG Life Science Institute (https://www.biolog.de). [a] The number assigned to the $NAD^+$ analog for this study. [b] Abbreviation assigned by BIOLOG Life Science Institute and used herein. For the clickable analogs: *alkyne* = contains an alkyne group, or *azide* = contains an azide group for use in copper-catalyzed alkyne-azide cycloaddition ('click') reactions. [c] BIOLOG Life Science Institute catalog number.

• Supplementary file 3. PARP-7 protein substrates from OVCAR4 and HeLa cells identified using an asPARP-7 approach with 8-Bu(3-yne)T-$NAD^+$. Cytosolic extracts prepared from OVCAR4 and HeLa cells were incubated with purified asPARP-7 S563G and 8-Bu(3-yne)T-$NAD^+$. The 8-Bu(3-yne)T-ADP-ribosylated proteins from the reactions were covalently linked to azido-agarose beads via copper-catalyzed cycloaddition. The conjugated azido-agarose beads were washed extensive and trypsinized to release peptides for protein identification by LC-MS/MS analysis. The 'Column Heading' key provides annotation information describing the metrics of each of the LC-MS/MS-identified peptides. The 'Tab' key provides details about all of the other worksheets contained within this spreadsheet.

• Supplementary file 4. Sequences of tryptic peptides from the asPARP-7 protein identifications. Sequences of the trypsinized peptides from the asPARP-7 protein identifications shown in *Supplementary file 3*.

• Supplementary file 5. Identification of PARP-7-mediated sites of ADPRylation on α-tubulin using the $NAD^+$ analog-sensitive approach. OVCAR4 and HeLa cell cytosolic extracts were incubated with recombinant analog sensitive PARP-7 (asPARP-7) in the presence of 8-Bu(3-yne)T-NAD+. Following in vitro modification, the extract proteins were covalently linked to azido-agarose beads via copper-catalyzed cycloaddition. The conjugated beads were washed, trypsinized to release peptides for protein identification (*Supplementary files 3* and *4*), and then washed again. The remaining peptides containing ADP-ribosylation sites were eluted from the resin using hydroxylamine (NH2OH). The cleaved modification produces a 15.019 m/z shift identifying the specific site of glutamate or

aspartate modification. Both the tryptic digest (PeptideID) and hydroxylamine eluate (SiteID) were subjected to LC-MS/MS analysis. The 'Summary' worksheet in this spreadsheet provides a summary of all data identifying sites of PARP-7-mediated ADPRylation on α-tubulin. The 'Data' worksheet provides mass spec data and annotation information describing the metrics of each of the LC-MS/MS-identified peptides. All analyses and numbering of amino acids are for human α-tubulin.

- Transparent reporting form

## Data availability

The RNA-seq sets generated for this study can be accessed from the NCBI's Gene Expression Omnibus (GEO) repository (http://www.ncbi.nlm.nih.gov/geo/) using the superseries accession number GSE153395. The new mass spec data sets generated for these studies are available as supplemental data provided with this manuscript. They can also be accessed from the Spectrometry Interactive Virtual Environment (MassIVE) repository (https://massive.ucsd.edu/ProteoSAFe/static/massive.jsp) using accession number MSV000086611.

The following datasets were generated:

| Author(s) | Year | Dataset title | Dataset URL | Database and Identifier |
|---|---|---|---|---|
| Parsons LHP, Gibson BA, Challa S, Nandu T., Stokes MS, Huang D, Lea JS, Kraus WL | 2021 | Substrate Identification Using a Chemical Genetics Approach Reveals a Role for PARP-7-Mediated MARylation in Controlling Microtubule Stability in Ovarian Cancer Cells | https://www.ncbi.nlm.nih.gov/geo/query/acc.cgi?acc=GSE153395 | NCBI Gene Expression Omnibus, GSE153395 |
| Huang D, Lea JS, Kraus WL, Parsons LHP, Gibson BA, Challa S, Nandu T., Stokes MS | 2021 | Substrate Identification Using a Chemical Genetics Approach Reveals a Role for PARP-7-Mediated MARylation in Controlling Microtubule Stability in Ovarian Cancer Cells | https://doi.org/10.25345/C5KN3G | MassIVE, 10.25345/C5KN3G |

The following previously published datasets were used:

| Author(s) | Year | Dataset title | Dataset URL | Database and Identifier |
|---|---|---|---|---|
| The GTEx Consortium | 2020 | GTEx Analysis Release V8 | https://www.ncbi.nlm.nih.gov/projects/gap/cgi-bin/study.cgi?study_id=phs000424.v8.p2 | dbGaP, phs000424.v8.p2 |

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
