## [Decision Letter]

**Acceptance summary:**

Using chemical genetics approaches, this study along with the back-to-back study from the Cohen group represent the first systematic investigations on PARP-7 regulated proteome. Particularly, this study not only identifies PARP-7 substrates but also dissects the biological roles of α-tubulin ADP-ribosylation and PARP-7 catalytic activity in cancer phenotypes. Intriguingly, this study identifies part of the PARP-7 specificity being towards the acidic residues, in contrast to the cysetine specificity in the Cohen study. Given the recent findings on the role of PARP-7 in viral infection and the development of PARP-7 inhibitors for cancer therapy, this work is timely for the field and beyond.

**Decision letter after peer review:**

Thank you for submitting your article "Identification of PARP-7 substrates reveals a role for MARylation in microtubule control in ovarian cancer cells" for consideration by *eLife*. Your article has been reviewed by three peer reviewers, including Anthony K L Leung as the Reviewing Editor and Reviewer #1, and the evaluation has been overseen by Philip Cole as the Senior Editor. The following individual involved in review of your submission have agreed to reveal their identity: and Ivan Matic (Reviewer #2).

The reviewers have discussed the reviews with one another and the Reviewing Editor has drafted this decision to help you prepare a revised submission.

As the editors have judged that your manuscript is of interest, but as described below that additional experiments are required before it is published, we would like to draw your attention to changes in our revision policy that we have made in response to COVID-19 (https://elifesciences.org/articles/57162). First, because many researchers have temporarily lost access to the labs, we will give authors as much time as they need to submit revised manuscripts. We are also offering, if you choose, to post the manuscript to bioRxiv (if it is not already there) along with this decision letter and a formal designation that the manuscript is "in revision at *eLife*".

Summary:

Parsons, Gibson, Challa et al. created a new variant of their chemical genetics methods to identify PARP7 targets. The authors identified two PARP7 gatekeeper residues and generated 10 different mutants, which were screened for their catalytic activity in the presence of four different NAD analogs. The authors used S563G mutant with the 8-Bu(3-yne)T-NAD+ analog to identify PARP7 substrates by mass spectrometry, which represents an important resource of PARP7 targets that can be explored in future studies. In addition, the authors reported the mRNA expression of PARP7 is lower in cancer than with normal tissue. Among the identified targets, they focus on α-tubulin, indicating that PARP7 is important for the maintenance of microtubules dynamics, suggesting a possible mechanistic bridge between ovarian cancer growth and PARP7 activity. Given that PARP7 inhibitor is now in Phase I clinical trials for solid tumors, this manuscript represents a significant advance and should be considered for publication with the following improvements:

Essential revisions:

1) Extend the mechanistic insights on PARP7 and microtubules: "PARP-7 mediated microtubule control may play a role in the regulation of cancer cell growth and motility" This statement is premature without further experimental data. The mechanism of the observed PARP7-dependent change in α-tubulin in two tubulin-sensitized conditions (cold and nocodazole treatment) is unclear. Would the level of α-tubulin and its ADP-ribosylation change at the basal, stressed (cold/nocodazole) and recovered in wild-type and PARP7 knockdown condition? Along the same lines, Figure 7C lacks immunofluorescence data of WT and PARP7 KD cells not treated with cold or nocodazole, which is an important control to aid in interpretation of the results.

How does this observed change in α-tubulin status relate to cancer phenotype? It is hard to reconcile the observations that PARP7 depletion stabilizes microtubule structure with the decreased cell growth, migration, and invasion observed upon PARP7 knockdown. Is the ability to quickly polymerize and assemble microtubules important for efficient cell growth and migration in cancer cells? If so, it might indicate that the observed phenotype in this cell line is not dependent upon PARP7 ADP-ribosylation of α-tubulin, and may be partially hampered by it. Providing in the text a more thorough exposition of the consequences of α-tubulin dynamics and dysregulation in ovarian cancer may help resolve this apparent contradiction, especially for readers not familiar with the field.

Lastly, without identifying α-tubulin residues that are MARylated by PARP7 and showing that their mutation has an impact on miroctubule stability, the authors cannot claim, at this stage, a direct link between PARP7 and microtubules. Therefore, the authors may need to limit their claims in the Title, Abstract and Discussion.

2) Is the observed effect due to PARP7 catalytic activity? Although PARP7 knockdown resulted in less MARylation on α-tubulin, it is unclear whether the effect is direct or indirect. Can the author rescue the ADP-ribosylation of α-tubulin by wild-type PARP7 but not catalytically inactive PARP7 in the PARP7 knockdown cells? Would the effects identified in α-tubulin be reproducible using the PARP7-specific inhibitor such as RBN-2397? Would the inhibitor reduce proliferation in ovarian cancer cells through cell growth and motility?

3) PARP7 expression in ovarian cancer cell and oncogenesis: Figure 1 legend states that expression levels across tissues were normalized under B) but not A). Could the authors describe how the data were normalized under B) and could they also show normalized data under A). Would the use of GEPIA as mentioned in the Materials and methods be sufficient in this normalization? Do the authors have data to support the case? TPMs presume that the total RNA content between the tissues is the same. This is, however, often not the case. Therefore, it would be good to normalize the data using housekeeping genes. The following article provides a list of genes that have been found to have stable expression over human tissues: https://www.cell.com/trends/genetics/pdf/S0168-9525(13)00089-9.pdf

Gene expression levels in an ovarian cancer cell line such as OVCAR4 can be very different compared to ovarian cancer tissue. Therefore, PARP7 expression levels should be compared between non-ovarian versus ovarian cells and ovarian normal vs ovarian cancer cells. As mentioned in the Discussion, it is puzzling how PARP7 expression levels are reduced in ovarian cancer tissue, which would suggest a tumor suppressor role for PARP7, and yet PARP7 silencing reduces proliferation and migration properties of OVCAR4 cells as reported in this manuscript, suggesting that it acts as an oncogene. This conundrum could be due to different PARP7 expression levels in ovarian tissue versus ovarian cell lines or specifically the ovarian cancer cell line OVCAR4 used in this study. One issue with the current analysis is that the authors are not comparing PARP7 expression in normal vs ovarian cancer cells. Instead they are comparing tissues, which consists of different cell types (stromal cells, fibroblasts, immune cells, etc.). In this case, the authors may want to check the expression of PARP7 in individual ovarian cell types from single cell data:

Healthy: https://www.ncbi.nlm.nih.gov/pmc/articles/PMC7052271/

Ovarian cancer: https://www.nature.com/articles/s41591-020-0926-0

If properly analyzed, these data should elucidate how PARP7 behaves during carcinogenesis.

4) Would the observed effect be restricted to OVCAR4 (e.g., α-tubulin effects and reduction proliferation due to PARP7 knockdown)? The reported PARP7-dependent ADPr of α-tubulin, and its role as mediator of microtubule dynamics in ovarian cancer cells is intriguing, but it is not clear if this role is specific for ovarian cancer cell lines or a general property of PARP7-expressing cells. For successful translation of these results into a clinically-relevant settings, an important step is to clarify this point by comparing the α-tubulin behaviour of OVCAR4 cell line to that of other cell lines. Similarly, the authors show that cell growth of OVCAR4 with PARP7 knockdown is impaired compared to WT, but how does PARP7 knockdown affect other highly-proliferative cell lines?

---

## [Author Response]

Essential revisions:1) Extend the mechanistic insights on PARP7 and microtubules: "PARP-7 mediated microtubule control may play a role in the regulation of cancer cell growth and motility" This statement is premature without further experimental data. The mechanism of the observed PARP7-dependent change in α-tubulin in two tubulin-sensitized conditions (cold and nocodazole treatment) is unclear. Would the level of α-tubulin and its ADP-ribosylation change at the basal, stressed (cold/nocodazole) and recovered in wild-type and PARP7 knockdown condition? Along the same lines, Figure 7C lacks immunofluorescence data of WT and PARP7 KD cells not treated with cold or nocodazole, which is an important control to aid in interpretation of the results.

We thank the reviewer for giving us an opportunity to address this important point. We have added new data showing that recovery from tubulin sensitizing treatments, such as nocodazole or cold, leads to enhanced expression of PARP-7 and increased MARylation of ⍺-tubulin (new Figure 7D). Together with our previous observation that PARP-7 regulates ⍺-tubulin MARylation, our results identify a novel regulatory pathway of ⍺-tubulin dynamics in which PARP-7 upregulation enhances ⍺-tubulin MARylation when the cells recover from tubulin stressors.

More specifically, to address the reviewer’s concern, we added new experiments examining the ADPRylation of α-tubulin and microtubule stability under an additional set of conditions:

1) α-tubulin MARylation under basal conditions and after nocodazole or cold treatment (new Figure 7D). These results show that the levels of α-tubulin MARylation and PARP-7 increase after treatment with the microtubule depolymerizing agents.

2) α-tubulin ADPRylation and microtubule stability under basal conditions and after nocodazole or cold treatment, with or without *PARP7* KD ± complementation with Wt or catalytically dead PARP7 (new Figure 7—figure supplement 2A and B). These results support the conclusion that PARP-7, through its catalytic activity, ADPRylates α-tubulin to destabilize microtubules.

3) We have added immunofluorescence data of Wt and *PARP7* KD cells not treated with cold or nocodazole (i.e., the basal condition) to Figure 7C, as requested. Interestingly, these control conditions suggest that PARP-7 catalytic activity is required for the microtubule destabilizing effects of nocodazole or cold treatment.

Taken together, our new and previous results show a clear role for PARP-7-mediated α-tubulin ADPRylation in nocodazole- or cold-induced microtubule instability.

How does this observed change in α-tubulin status relate to cancer phenotype? It is hard to reconcile the observations that PARP7 depletion stabilizes microtubule structure with the decreased cell growth, migration, and invasion observed upon PARP7 knockdown. Is the ability to quickly polymerize and assemble microtubules important for efficient cell growth and migration in cancer cells? If so, it might indicate that the observed phenotype in this cell line is not dependent upon PARP7 ADP-ribosylation of α-tubulin, and may be partially hampered by it. Providing in the text a more thorough exposition of the consequences of α-tubulin dynamics and dysregulation in ovarian cancer may help resolve this apparent contradiction, especially for readers not familiar with the field.

We appreciate the reviewer’s concern and agree that we could have done a better job putting these results in a biological context. As the reviewers noted, at face value, our results suggest that enhanced microtubule stability upon depletion of PARP-7 or loss of its catalytic activity has negative consequences for cancer phenotypes (i.e., it’s bad for the cancer cells). This suggests that in ovarian cancer cells, PARP-7 acts to MARylate α-tubulin to destabilize microtubules. How does this help cancer cells?

One possibility, as the reviewer suggested, is that the ability to quickly depolymerize and disassemble microtubules is important for efficient cell growth and migration. Indeed, there is evidence for this in the literature. In fact, there is a class of anticancer drugs that stabilize microtubules (e.g., taxanes), leading to the arrest of proliferation and mitosis. Moreover, previous studies have shown that well characterized tumor suppressors such as RASSF1A and APC act to stabilize microtubule polymerization (PMIDs: 14603253 and 11237529). We have now addressed more clearly the implications of our results for cancer biology and therapeutics in the Discussion.

As described in our response below, our new results with an α-tubulin MARylation site mutant do indeed indicate that the observed phenotype is dependent upon PARP-7 MARylation of α-tubulin.

Lastly, without identifying α-tubulin residues that are MARylated by PARP7 and showing that their mutation has an impact on miroctubule stability, the authors cannot claim, at this stage, a direct link between PARP7 and microtubules. Therefore, the authors may need to limit their claims in the Title, Abstract and Discussion.

The reviewer raises a good point. To strengthen our conclusions, we have identified the sites of PARP-7-dependent MARylation on α-tubulin, mutated them, and performed functional analyses with the mutant (new Figure 7E-G). Our analysis identified three sites of MARylation: D69, E71, E77. We mutated each of the residues to similar, but non-modifiable residues (D à N and E à Q), generated expression vectors, and expressed the wild-type or mutant versions of α-tubulin in OVCAR4 cells. Mutation of these sites inhibited MARylation of α-tubulin (Figure 7E). Importantly, expression of the α-tubulin MARylation site mutant phenocopied *PARP7* knockdown (Figure 7C) and expression of a catalytically dead PARP-7 mutant (new Figure 7—figure supplement 2B) in the microtubule stability assay (new Figure 7F and G). These results show that MARylation of α-tubulin is required for the observed effects of PARP-7.

2) Is the observed effect due to PARP7 catalytic activity? Although PARP7 knockdown resulted in less MARylation on α-tubulin, it is unclear whether the effect is direct or indirect. Can the author rescue the ADP-ribosylation of α-tubulin by wild-type PARP7 but not catalytically inactive PARP7 in the PARP7 knockdown cells? Would the effects identified in α-tubulin be reproducible using the PARP7-specific inhibitor such as RBN-2397? Would the inhibitor reduce proliferation in ovarian cancer cells through cell growth and motility?

We thank the reviewer for this important comment. We performed additional experiments to test whether PARP-7 catalytic activity is required for the observed phenotype. We used a catalytically dead PARP-7 mutant (Y564A) suggested by the reviewer (new Figure 7—figure supplement 2). We expressed the mutant from an Flag-tagged mouse *Parp7* cDNA to produce an shRNA-resistant mRNA that could be used in knockdown-addback experiments (the *PARP7* shRNAs that we used were designed against the human *PARP7* mRNA). Expression of the catalytically dead PARP-7 mutant resulted in a loss of α-tubulin MARylation (new Figure 7—figure supplement 2A). In addition, expression of the catalytically dead PARP-7 mutant phenocopied *PARP7* knockdown and expression of the α-tubulin ADPRylation site mutant in the microtubule stability assay (new Figure 7—figure supplement 2B) and the cell migration assay (new Figure 7—figure supplement 2C and D) (i.e., it was impaired versus wild-type PARP-7 in both cases). These results show that PARP-7 catalytic activity is required for the observed effects of PARP-7. We preferred this approach to the use of a chemical inhibitor of PARP-7, such as RBN2397, to eliminate potential off target effects and uncertainty of proper dosing in OVCAR4 cells.

3) PARP7 expression in ovarian cancer cell and oncogenesis: Figure 1 legend states that expression levels across tissues were normalized under B) but not A). Could the authors describe how the data were normalized under B) and could they also show normalized data under A). Would the use of GEPIA as mentioned in the Materials and methods be sufficient in this normalization? Do the authors have data to support the case? TPMs presume that the total RNA content between the tissues is the same. This is, however, often not the case. Therefore, it would be good to normalize the data using housekeeping genes. The following article provides a list of genes that have been found to have stable expression over human tissues: https://www.cell.com/trends/genetics/pdf/S0168-9525(13)00089-9.pdf

We appreciate the concerns raised by the reviewer, which we have addressed by clarifying the text and adding new analyses. First, we have edited the text to remove the impression that the analysis in Figure 1A was not normalized; this is not true. The GTEx data in Figure 1A were obtained from the GTEx portal and used as normalized TPM scores as described by Li et al., 2010. We have clarified this in the text. Second, we normalized the GTEx data in Figure 1A to 50 housekeeping genes, as suggested by the reviewer (new Figure 1—figure supplement 1A). This normalization did not fundamentally change the results.

Gene expression levels in an ovarian cancer cell line such as OVCAR4 can be very different compared to ovarian cancer tissue. Therefore, PARP7 expression levels should be compared between non-ovarian versus ovarian cells and ovarian normal vs ovarian cancer cells. As mentioned in the Discussion, it is puzzling how PARP7 expression levels are reduced in ovarian cancer tissue, which would suggest a tumor suppressor role for PARP7, and yet PARP7 silencing reduces proliferation and migration properties of OVCAR4 cells as reported in this manuscript, suggesting that it acts as an oncogene. This conundrum could be due to different PARP7 expression levels in ovarian tissue versus ovarian cell lines or specifically the ovarian cancer cell line OVCAR4 used in this study. One issue with the current analysis is that the authors are not comparing PARP7 expression in normal vs ovarian cancer cells. Instead they are comparing tissues, which consists of different cell types (stromal cells, fibroblasts, immune cells, etc.). In this case, the authors may want to check the expression of PARP7 in individual ovarian cell types from single cell data:Healthy: https://www.ncbi.nlm.nih.gov/pmc/articles/PMC7052271/Ovarian cancer: https://www.nature.com/articles/s41591-020-0926-0If properly analyzed, these data should elucidate how PARP7 behaves during carcinogenesis.

The reviewer has raised a very good point. Our new analyses, as well as additional information about the origins of ovarian cancer from the literature, suggest that the mixed cell populations in normal and cancerous ovarian tissues are a complicating factor when analyzing and comparing *PARP7* expression. We used the scRNA-seq data suggested by the reviewer to analyze *PARP7* expression in normal and cancerous ovarian tissues at the single-cell level (new Figure 1—figure supplement 1B). We observed that the identifiable cell types in normal and cancerous ovarian tissues are quite different. The main cell types identifiable as “ovarian” in normal ovary (oocyte, granulosa, stroma) are not well represented in ovarian cancer. Thus, as the reviewer noted, it is difficult to compare *PARP7* expression in the two tissues. By these metrics, *PARP7* expression in the “ovarian” cell types varies quite a bit, and may actually be lower on average than in the “malignant” ovarian cancer cell types (new Figure 1—figure supplement 1B). In this regard, *PARP7* gene gain and amplification is a common feature of ovarian cancers (new Figure 1—figure supplement 1C).

Furthermore, when contemplating these results, we were drawn to a growing body of work indicating that ovarian cancers may actually arise from cells in the Fallopian tubes. Thus, attempts to compare *PARP7* expression in normal and cancerous ovarian tissues may be like comparing apples and oranges: (1) the cellular origins of the tissues are different and (2) the cellular compositions are different. Thus, while the results presented in Figure 1B are correct, the require some context and qualification. We have now provided this through (1) new Figure 1—figure supplement 1A and B, (2) presentation of these data in the appropriate section of the Results, and (3) discussion of the results in the Discussion.

4) Would the observed effect be restricted to OVCAR4 (e.g., α-tubulin effects and reduction proliferation due to PARP7 knockdown)? The reported PARP7-dependent ADPr of α-tubulin, and its role as mediator of microtubule dynamics in ovarian cancer cells is intriguing, but it is not clear if this role is specific for ovarian cancer cell lines or a general property of PARP7-expressing cells. For successful translation of these results into a clinically-relevant settings, an important step is to clarify this point by comparing the α-tubulin behaviour of OVCAR4 cell line to that of other cell lines. Similarly, the authors show that cell growth of OVCAR4 with PARP7 knockdown is impaired compared to WT, but how does PARP7 knockdown affect other highly-proliferative cell lines?

To address the reviewer’s concern, we performed key cancer phenotypic assays (i.e., growth, migration, invasion) upon *PARP7* knockdown in three additional cancer cell lines: HeLa (cervical cancer), OVCAR3 (ovarian cancer), A704 (kidney cancer), whose PARP-7 levels are comparable to OVCAR4 cells. In all three cell lines and all three assays, *PARP7* knockdown caused an inhibition of the cancer-related phenotypes (new Figure 1—figure supplement 2A-L) and microtubule dynamics (new Figure 7—figure supplement 1A-C) similar to what we observed in OVCAR4 cells. Thus, the observed effects of PARP-7 on cancer-related endpoints are not restricted to OVCAR4 cells.